# A gene desert required for regulatory control of pleiotropic *Shox2* expression and embryonic survival

Samuel Abassah-Oppong [1,13,15], Matteo Zoia[2,15], Brandon J. Mannion[3,4,15], Raquel Rouco [5], Virginie Tissières[2,6,7], Cailyn H. Spurrell [3], Virginia Roland[2], Fabrice Darbellay [3,5], Anja Itum[1], Julie Gamart[2,7], Tabitha A. Festa-Daroux[1], Carly S. Sullivan[1], Michael Kosicki[3], Eddie Rodríguez-Carballo[8,14], Yoko Fukuda-Yuzawa[3], Riana D. Hunter[3], Catherine S. Novak[3], Ingrid Plajzer-Frick[3], Stella Tran[3], Jennifer A. Akiyama [3], Diane E. Dickel[3], Javier Lopez-Rios [6,9], Iros Barozzi [3,10], Guillaume Andrey [5], Axel Visel [3,11,12], Len A. Pennacchio [3,4,11], John Cobb [1] ✉ & Marco Osterwalder [2,3,7] ✉

Approximately a quarter of the human genome consists of gene deserts, large regions devoid of genes often located adjacent to developmental genes and thought to contribute to their regulation. However, defining the regulatory functions embedded within these deserts is challenging due to their large size. Here, we explore the *cis*-regulatory architecture of a gene desert flanking the *Shox2* gene, which encodes a transcription factor indispensable for proximal limb, craniofacial, and cardiac pacemaker development. We identify the gene desert as a regulatory hub containing more than 15 distinct enhancers recapitulating anatomical subdomains of *Shox2* expression. Ablation of the gene desert leads to embryonic lethality due to *Shox2* depletion in the cardiac sinus venosus, caused in part by the loss of a specific distal enhancer. The gene desert is also required for stylopod morphogenesis, mediated via distributed proximal limb enhancers. In summary, our study establishes a multi-layered role of the *Shox2* gene desert in orchestrating pleiotropic developmental expression through modular arrangement and coordinated dynamics of tissue-specific enhancers.

Functional assessment of gene deserts, gene-free chromosomal segments larger than 500 kilobases (kb), has posed considerable challenges since these large noncoding regions were shown to be a prominent feature of the human genome more than 20 years ago[1]. Stable gene deserts (*n* = 172 in the human genome, ~30% of all gene deserts) share more than 2% genomic sequence conservation between human and chicken, are enriched for putative enhancer elements and frequently located near developmental genes, suggesting a critical role in embryonic development and organogenesis[2–4]. However, genomic deletion of an initially selected pair of gene deserts displayed mild effects on the expression of nearby genes and absence of overt phenotypic alterations[5]. In contrast, gene deserts centromeric and telomeric to the *HoxD* cluster were shown to harbor "regulatory archipelagos" i.e., multiple tissue-specific enhancers that collectively orchestrate spatiotemporal and colinear *HoxD gene* expression in developing limbs and other embryonic compartments[6,7]. These antagonistic gene deserts represent individual topologically associating domains (TADs) separated by the *HoxD* cluster which acts as a dynamic and resilient CTCF-enriched boundary region[8,9]. Despite such critical roles, the functional requirements of only few gene deserts

have been studied in detail, including the investigation of chromatin topology and functional enhancer landscapes in the TADs of other key developmental transcription factors (TFs), such as *Sox9* and *Hoxa2*[10,11], or signaling ligands, such as *Shh* and *Fgf8*[12,13].

Self-associating TADs identified by 3D chromatin conformation capture are described as primary higher-order chromatin structures that constrain *cis*-regulatory interactions to target genes and facilitate dynamic long-range enhancer-promoter (E-P) contacts[14–16]. TADs are thought to emerge through Cohesin-mediated chromatin loop extrusion and are delimited by association of CTCF to convergent binding sites[17,18]. Re-distribution of E-P interactions can lead to pathogenic effects due to perturbation of CTCF-bound TAD boundaries or re-configuration of TADs[10,19]. Therefore, functional characterization of the 3D chromatin topology and transcriptional enhancer landscapes across gene deserts is a prerequisite for understanding the developmental mechanisms underlying mammalian embryogenesis and human syndromes[20]. Recent functional studies in mice have uncovered that mRNA expression levels of developmental regulator genes frequently depend on additive contributions of enhancers within TADs[21–24]. Hereby, the contribution of each implicated enhancer to total gene dosage can vary, illustrating the complexity of transcriptional regulation through E-P interactions[25]. In addition, nucleotide mutations affecting TF binding sites in enhancers can disturb spatio-temporal gene expression patterns, with the potential to trigger phenotypic abnormalities such as congenital malformations due to altered properties of developmental cell populations[26–28].

In this study, we focused on the functional characterization of a stable gene desert downstream (centromeric) of the mouse short stature homeobox 2 (*Shox2*) transcription factor (TF) gene. Tightly controlled *Shox2* expression is essential for accurate development of the stylopod (humerus and femur), craniofacial compartments (maxillary-mandibular joint, secondary palate), the facial motor nucleus and its associated facial nerves, and a subset of neurons of the dorsal root ganglia[29–35]. In addition, *Shox2* in the cardiac sinus venosus (SV) is required for differentiation of progenitors of the sinoatrial node (SAN), the dominant pacemaker population during embryogenesis and adulthood[36–38]. *Shox2* inactivation disrupts Nkx2-5 antagonism in SAN pacemaker progenitors and results in hypoplasia of the SAN and venous valves, leading to bradycardia and embryonic lethality[36,37,39]. In accordance with this role, *SHOX2*-associated coding and non-coding variants in humans were implicated with SAN dysfunction and atrial fibrillation[40–42]. The *Tbx5* and *Isl1* TF genes were shown to act upstream of *Shox2* in SAN development[43–46] and *Isl1* is sufficient to rescue *Shox2*-mediated bradycardia in zebrafish hearts[47].

In humans, the *SHOX* gene located on the pseudo-autosomal region (PAR1) of the X and Y chromosomes represents a paralog of *SHOX2* (on chromosome 3), hence *Shox* gene function is divided. *SHOX* is associated with defects and syndromes affecting skeletal, limb and craniofacial morphogenesis[30,48,49]. Rodents have lost their *SHOX* gene during evolution along with other pseudo-autosomal genes and mouse *Shox2* features an identical DNA-interacting homeodomain replaceable by human *SHOX* in a mouse knock-in model[29,50]. Remarkably, while *Shox2/SHOX2* genes show highly conserved locus architecture, the *SHOX* gene also features a downstream gene desert of similar extension, containing neural (hindbrain) enhancers with overlapping activities[49]. Our previous studies revealed that *Shox2* transcription in the developing mouse stylopod is partially controlled by a pair of human-conserved limb enhancers termed hs741 and hs1262/LHB-A, the latter residing in the gene desert[21,49,51]. However, the rather moderate loss of *Shox2* limb expression in absence of these enhancers indicated increased complexity and potential redundancies in the underlying enhancer landscape[21].

Here we identified the *Shox2* gene desert as a critical *cis*-regulatory domain encoding an array of distal enhancers with specific subregional activities, predominantly in limb, craniofacial, neuronal,

and cardiac cell populations. We found that interaction of these enhancers with the *Shox2* promoter is likely facilitated by a chromatin loop anchored downstream of the *Shox2* gene body and exhibiting tissue-specific features. Genome editing further demonstrated essential pleiotropic functions of the gene desert, including a requirement for craniofacial patterning, limb morphogenesis, and embryonic viability through enhancer-mediated control of SAN progenitor specification. Our results identify the *Shox2* gene desert as a dynamic enhancer hub ensuring pleiotropic and resilient *Shox2* expression as an essential component of the gene regulatory networks (GRNs) orchestrating mammalian development.

## Results
### Gene desert enhancers recapitulate patterns of pleiotropic *Shox2* expression

The gene encoding the SHOX2 transcriptional regulator is located in a 1 megabase (Mb) TAD (chr3:66337001-67337000) and flanked by a stable gene desert spanning 675 kb of downstream (centromeric) genomic sequence (Fig. 1A). The *Shox2* TAD only contains one other protein coding gene, *Rsrc1*, located adjacent to *Shox2* on the upstream (telomeric) side and known for roles in pre-mRNA splicing and neuronal transcription[52,53] (Fig. 1A). Genes located beyond the TAD boundaries show either near-ubiquitous (*Mlf1*) or *Shox2*-divergent (*Veph1*, *Ptx3*) expression signatures across tissues and timepoints (Supplementary Fig. 1A). While *Shox2* transcription is dynamically regulated in multiple tissues including proximal limbs, craniofacial subregions, cranial nerve, brain, and the cardiac sinus venosus (SV), only a limited number of *Shox2*-associated enhancer sequences have been previously validated in mouse embryos[21,49,51,54] (Fig. 1A, B, Supplementary Fig. 1A). These studies identified a handful of conserved human (hs) and mouse (mm) enhancer elements in the *Shox2* TAD driving reporter activity almost exclusively in the mouse embryonic brain (hs1413, hs1251, hs1262) and limbs (hs741, hs1262, hs638/mm2107) (Vista Enhancer Browser) (Fig. 1A). In addition, a recent study identified a human enhancer sequence (termed R4) that drove activity in the SV[55]. To predict *Shox2* enhancers more systematically, and to estimate the number of developmental enhancers in the gene desert, we established a map of stringent enhancer activities based on chromatin state profiles[56] (ChromHMM) and H3K27 acetylation (H3K27ac) ChIP-seq peak calls across 66 embryonic and perinatal tissue-stage combinations from ENCODE[57] (https://www.encodeproject.org) (see Methods). After excluding promoter regions, this analysis identified 20 elements within the *Shox2*-TAD and its border regions, each with robust enhancer marks in at least one of the tissues and timepoints (E11.5-15.5) examined (Supplementary Fig. 1A and Supplementary Data 1). Remarkably, 17 of the 20 elements mapping to the *Shox2* TAD or border regions were located within the downstream gene desert, with the majority of H3K27ac signatures overlapping *Shox2* expression profiles across multiple tissues and timepoints, indicating a role in regulation of pleiotropic *Shox2* expression (Fig. 1B, Supplementary Fig. 1A). The previously validated hs741 and hs1262 limb enhancers were not among the stringent predictions across timepoints as H3K27ac levels at these enhancers are progressively reduced after E10.5[58,59], despite strong LacZ reporter signal in the proximal limb at subsequent stages (Fig. 1A)[21,49]. Reducing stringency of H3K27ac-thresholds and including E10.5 profiles however extended the number of predicted enhancer elements within the *Shox2* TAD significantly (Supplementary Data 1).

To determine the in vivo activity patterns for each of the predicted gene desert enhancer (DE) elements from stringent predictions, we performed LacZ transgenic reporter analysis in mouse embryos at E11.5, a stage characterized by wide-spread and functionally relevant *Shox2* expression in multiple tissues[21,49] (Fig. 1B, C). This analysis included the validation of 16 genomic elements (DE + 329 kb and +331 kb were part of a single reporter construct) and revealed

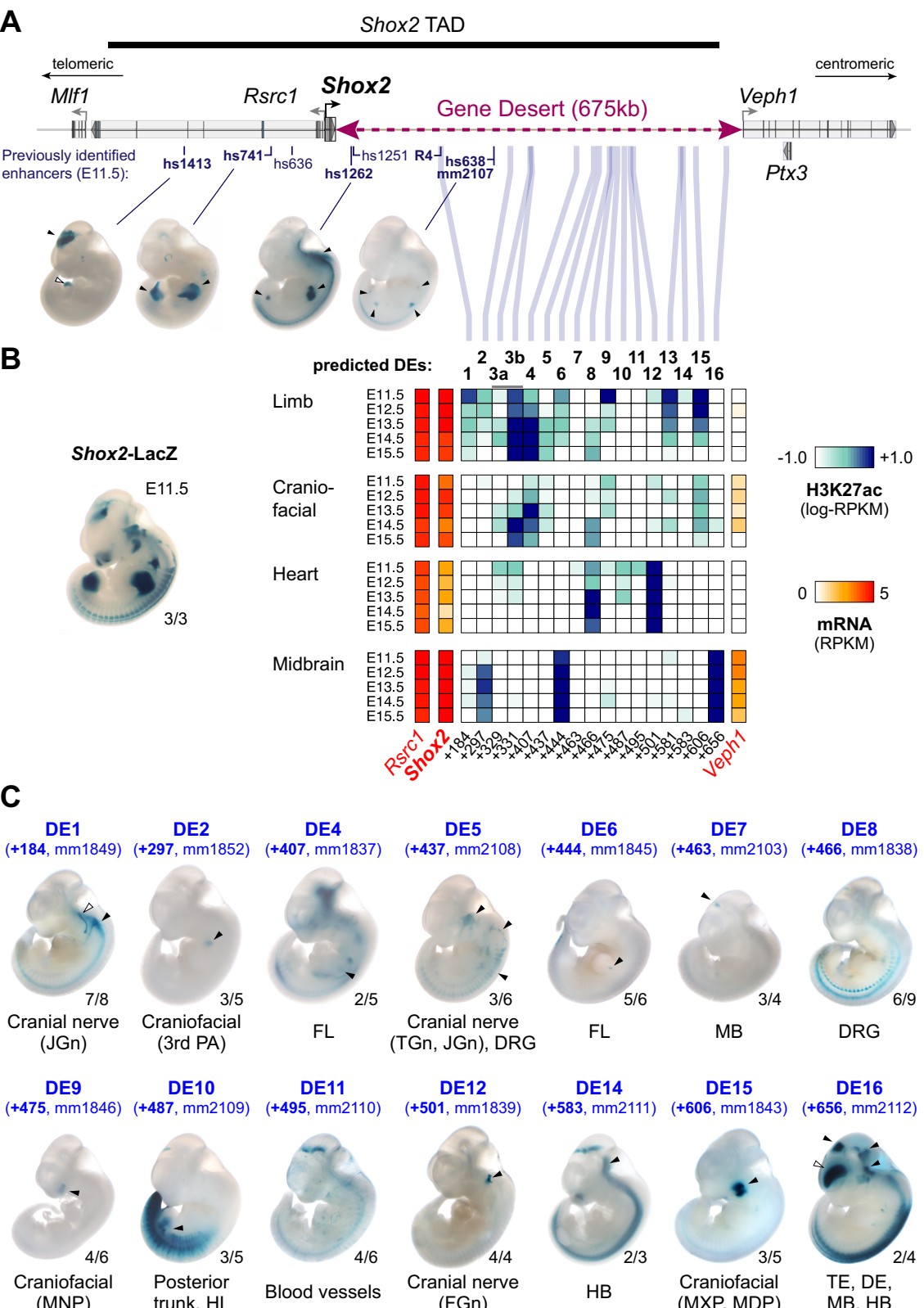

**Fig. 1**

reproducible enhancer activities in 14/16 cases (Fig. 1B, C, Supplementary Fig. 1B and Supplementary Table 1). Most of the individual enhancer activities localized to either craniofacial, cranial nerve, mid-/hindbrain or limb subregions known to be dependent on *Shox2* expression and function[29,30,32,33] (Fig. 1C). For example, DE9 (+475 kb) and DE15 (+606 kb), both exhibiting limb and craniofacial H3K27ac marks, drove LacZ reporter expression exclusively in *Shox2*-

overlapping craniofacial domains in the medial nasal (MNP) and maxillary-mandibular (MXP, MDP) processes, respectively (Fig. 1C). In line with DE15 activity, *Shox2* expression in the MXP-MDP junction is known to be required for temporomandibular joint (TMJ) formation in jaw morphogenesis[30]. DE1, 5 and 12 instead showed activities predominantly in cranial nerve tissue, including the trigeminal (TGn), facial (FGn) and jugular (JGn) ganglia, as well as the dorsal root ganglia

**Fig. 1 | The *Shox2* gene desert constitutes a hub for tissue-specific enhancers.**
**A** Genomic interval containing the *Shox2* TAD[64] and previously identified *Shox2*-associated enhancer regions. Vista Enhancer Browser IDs (hs: human sequence, mm: mouse sequence) in bold mark enhancers with *Shox2*-overlapping and reproducible activities (arrowheads). The position of the human R4 enhancer[55] driving reporter activity in the sinus venosus is indicated. **B** Heatmap showing H3K27 acetylation (ac) -predicted and ChromHMM-filtered putative enhancers and their temporal signatures in tissues with dominant *Shox2* functions (see full Supplementary Fig. 1). Blue and red shades represent H3K27ac enrichment and mRNA expression levels, respectively. Distance to *Shox2* TSS (+) is indicated in kb. Left: *Shox2* expression pattern (*Shox2*-LacZ/+) at E11.5[32]. **C** Transgenic LacZ reporter

validation of predicted gene desert enhancers (DEs) in mouse embryos at E11.5. Arrowheads point to reproducible enhancer activity with (black) or without (white) *Shox2* overlap. JGn, TGn, FGn: jugular, trigeminal, and facial ganglion, respectively. PA, pharyngeal arch. DRG, dorsal root ganglia. FL, Forelimb. HL, Hindlimb. TE, Telencephalon. DiE, Diencephalon. MB, Midbrain. HB, Hindbrain. MNP, medial nasal process. MXP, maxillary process. MDP, mandibular process. Reproducibility numbers are indicated on the bottom right of each representative embryo shown (reproducible tissue-specific staining vs. number of transgenic embryos with any LacZ signal). Corresponding Vista IDs of the elements tested are listed in Supplementary Table 1.

(DRG) (Fig. 1C). *Shox2* is expressed in all these neural crest-derived tissues, but a functional requirement has only been demonstrated for FGn development and the mechanosensory neurons of the DRG[32,34]. While no H3K27ac profiles for cranial nerve populations were available from ENCODE[58], both DE5 and DE12 elements showed increased H3K27ac in craniofacial compartments at E11.5 (Fig. 1B), likely reflecting the common neural-crest origin of a subset of these cell populations[60]. DE1, while representing the R4 mouse ortholog, did not reveal reproducible LacZ reporter activity in the heart at E11.5 (Fig. 1C, Vista Enhancer Browser). At mid-gestation, *Shox2* is also expressed in the diencephalon (DiE), midbrain (MB) and hindbrain (HB), and is specifically required for cerebellar development[33]. Gene desert enhancer assessment also identified a set of brain enhancers (DE7, 14 and 16) overlapping *Shox2* domains in the DiE, MB and/or HB (Fig. 1C). Although H3K27ac marks were present in limbs at most predicted DEs, only three elements (DE4, 6 and 10) drove LacZ reporter expression in the E11.5 limb mesenchyme in a sub-regionally or limb type-restricted manner (Fig. 1C). Remarkably, despite elevated cardiac H3K27ac in a subset of DEs, none of the validated elements drove reproducible LacZ reporter expression in the heart at E11.5 (Figs. 1B, C). DE3 ($n = 7$) and DE13 ($n = 5$) were the only elements not showing any reproducible activities in transgenic LacZ reporter embryos at E11.5. Taken together, our in vivo enhancer-reporter screen based on systematic epigenomic profiling and transgenic reporter validation identified multiple DE elements with *Shox2*-overlapping activities, pointing to a role of the gene desert as an enhancer hub directing pleiotropic *Shox2* transcription.

## The *Shox2* gene desert shapes a chromatin loop with tissue-specific features
Recent studies have shown that sub-TAD interactions can be pre-formed or dynamic, and that 3D chromatin topology can affect enhancer-promoter communication in distinct cell types or tissues[61–63]. To explore the 3D chromatin topology across the *Shox2* TAD and flanking regions, we performed region capture Hi-C (C-HiC) targeting a 3.5 Mb interval in dissected E11.5 mouse embryonic forelimbs, mandibles, and hearts, tissues known to be affected by *Shox2* loss-of-function (Fig. 2A, Supplementary Fig. 2A). C-HiC contact maps combined with analysis of insulation scores to infer inter-domain boundaries revealed a tissue-invariant *Shox2*-containing TAD that matched the extension observed in mESCs[64] (Fig. 2A, Supplementary Fig. 2A, Supplementary Data 2). C-HiC profiles further showed sub-TAD organization into *Shox2*-flanking upstream (U-dom) and downstream (D-dom) domains as hallmarked by loop anchors and insulation scores, with the D-dom spanning almost the entire gene desert (Fig. 2A, Supplementary Fig. 2A). Virtual 4 C (v4C) using a viewpoint centered on the *Shox2* transcriptional start site (TSS) further demonstrated confinement of *Shox2*-interacting elements to U-dom and D-dom intervals, or TAD boundary regions (Fig. 2A, Supplementary Data 2). Remarkably, the most distal D-dom compartment spanning ~170 kb revealed dense chromatin contacts restricted to heart tissue and delimited by weak insulation boundaries which co-localized with non-convergent CTCF sites (Fig. 2A, B, Supplementary Fig. 2A). While this high-density

contact domain (HCD) contained the majority of the previously identified (non-cardiac) gene desert enhancers (DE5-12), subtraction analysis further corroborated increased chromatin contacts across the HCD and domain insulation specifically in cardiac tissue as opposed to limb or mandibular tissue, potentially indicating a repressive function in heart cells due to condensed chromatin state (Fig. 2A–C, Supplementary Fig. 2A–C). However, no region-specific accumulation of repressive histone marks (H3K27me3 or H3K9me3, ENCODE) was observed in whole heart samples (Supplementary Fig. 3A, B). Instead, v4C subtraction analysis with defined viewpoints on positively validated DEs indicated that specifically in heart tissue, enhancer elements outside the HCD (DE1, 15) were reduced in contacts with elements inside (Supplementary Fig. 3C). In turn, enhancer viewpoints inside the HCD (DE5, 9, 10) showed reduced contacts with elements outside (Supplementary Fig. 3C). Collectively, our results imply that *Shox2* is preferentially regulated by upstream (U-dom) and downstream (D-dom) regulatory domains that contain distinct sets of active tissue-specific enhancers. Hereby, the gene desert forms a topological chromatin environment (D-dom) that in tissue-specific context might modulate the interaction of certain enhancers with the *Shox2* promoter.

## Control of pleiotropic *Shox2* dosage and embryonic survival by the gene desert
To explore the functional relevance of the gene desert as an interactive hub for *Shox2* enhancers in mouse embryos, we used CRISPR/Cas9 in mouse zygotes to engineer an intra-TAD gene desert deletion allele (GD$^\Delta$) (Fig. 3A, Supplementary Fig. 4A, B; Supplementary Tables 2, 3). F1 mice heterozygous for this allele (GD$^{\Delta/+}$) were born at expected Mendelian ratios and showed no impaired viability and fertility. Following intercross of GD$^{\Delta/+}$ heterozygotes we compared *Shox2* transcripts in GD$^{\Delta/\Delta}$ and wildtype (WT) control embryos, with a focus on tissues marked by DE activities (Figs. 1C, 3B–E). Despite loss of at least three enhancers with limb activities (hs1262, DE6, DE10), *Shox2* expression was still detected in fore- and hindlimbs of GD$^{\Delta/\Delta}$ embryos, as determined by in situ hybridization (ISH) (Fig. 3B), albeit at ~50% reduced transcript levels, as shown by qPCR (Fig. 3C, Supplementary Table 4). These results point to a functional role of the gene desert in ensuring robust *Shox2* dosage during proximal limb development[29]. Remarkably, *Shox2* expression in distinct craniofacial compartments was more severely affected by the loss of the gene desert (Fig. 3D, E). Downregulation of *Shox2* transcripts was evident in the MNP, anterior portion of the palatal shelves, and the proximal MXP-MDP domain of GD$^{\Delta/\Delta}$ embryos at E10.5 and E11.5, compared to wild-type controls (Fig. 3D). Concordantly, and in contrast to *Rsrc1* mRNA levels which remained unchanged, *Shox2* was depleted in the nasal process (NP) and MDP of GD$^{\Delta/\Delta}$ embryos at E11.5 (Fig. 3E). Strikingly, these affected subregions corresponded to the activity domains of the identified DE9 (MNP) and DE15 (MXP-MDP) elements indicating an essential requirement of these enhancers for craniofacial *Shox2* regulation (Figs. 1C, 3F). Taken together, these results demonstrate a critical functional role of the gene desert in transcriptional regulation of *Shox2* during cranio-facial and proximal limb morphogenesis[29,30,35]. While our transgenic

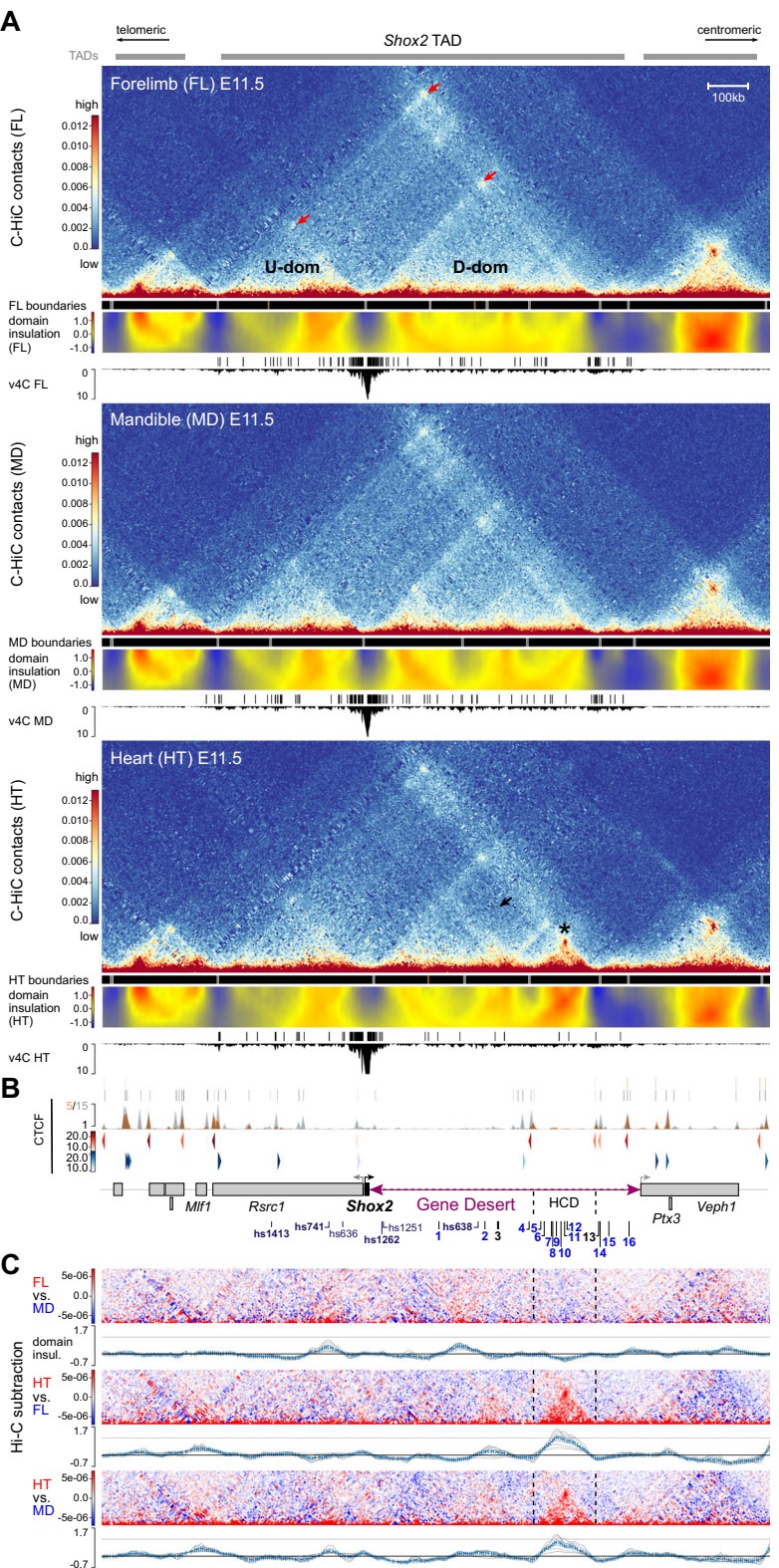

analysis also uncovered DEs with activities in brain or cranial nerve regions (Fig. 1C), no overt reduction in spatial *Shox2* expression was detected in corresponding subregions in GD^Δ/Δ embryos (Fig. 3D). This is likely attributed to the presence of *Shox2*-associated brain enhancers with partially overlapping activities and located in the U-dom (e.g., hs1413) or downstream of the deleted gene desert interval (e.g., DE16).

Despite the lack of identification of any in vivo heart enhancers in the gene desert following transgenic reporter analysis from

epigenomic whole-heart predictions (Fig. 1C), spatial and quantitative mRNA analysis in GD^Δ/Δ embryos revealed absence of *Shox2* transcripts from the cardiac sinus venosus (SV) that harbors the population of SAN pacemaker progenitors[65] (Fig. 4A, B). In accordance with the essential role of *Shox2* in the differentiation of SAN progenitors and the related lethality pattern in *Shox2*-deficient mouse embryos[36], cardiac *Shox2* depletion in GD^Δ/Δ embryos triggered arrested development and embryonic lethality at around E12 (*n* = 5/5) (Supplementary Fig. 4C).

**Fig. 2 | 3D chromatin architecture across the *Shox2* regulatory landscape in distinct tissues. A** C-HiC analysis of the genomic region containing the *Shox2* TAD[64] in wildtype mouse embryonic forelimb (FL), mandible (MD) and heart (HT) at E11.5 (see also Supplementary Fig. 2). The chr3:65977711-67631930 (mm10) interval is shown. Upper panels (for each tissue): Hi-C contact map revealing upstream (U-dom) and downstream (D-dom) domains flanking the *Shox2* gene. Middle panels: Stronger (gray boxes, $p < 0.01$) and weaker (brown boxes, $p > 0.01$, $<0.05$) domain boundaries based on TAD separation score (Wilcoxon rank-sum test). A matrix showing normalized inter-domain insulation score (blue = weak insulation, red = strong insulation) is plotted below. Bottom panels: Virtual 4 C (v4C) using a *Shox2*-centered viewpoint shows *Shox2* promoter interaction profiles in the different tissues. *Shox2* contacting regions ($q < 0.1$, Supplementary Data 2) as determined by GOTHiC[140] are shown on top. Red arrows point to chromatin domain anchors. Asterisk marks a high-density contact domain (HCD) observed only in heart tissue (chr3:66402500-66572500). Black arrow indicates reduction of internal D-dom contacts between elements inside the HCD and outside in the heart sample (see also Supplementary Fig. 2). **B** Top: CTCF enrichment in mESCs[64] (gray) and new-born mouse hearts at P0[58] (orange). Bottom: CTCF motif orientation (red/blue) and strength (gradient). Protein coding genes (gene bodies) are indicated below. DEs, predicted gene desert enhancers validated in Fig. 1 (blue: tissue-specific activity). **C** C-HiC subtraction to visualize tissue-specific contacts for each tissue comparison (red/blue). Plots below display the corresponding subtracted inter-domain insulation scores. Dashed lines demarcate the HCD borders.

---

Immunofluorescence further confirmed lack of SHOX2 protein in the HCN4-positive domain of SAN pacemaker cells in the SV of GD$^{\Delta/\Delta}$ hearts compared to WT controls at E11.5 (Fig. 4C, Supplementary Fig. 4D). Together, these results demonstrated a requirement of the gene desert for embryonic viability directly associated with transcriptional control of cardiac *Shox2*.

## Resilient gene desert enhancer architecture ensures robust cardiac *Shox2* expression

Abrogation of *Shox2* mRNA in the SV of GD$^{\Delta/\Delta}$ embryos implied the presence of enhancers with cardiac activities, similar to the regulation of other TFs implicated in the differentiation of SAN progenitor cells[46]. In agreement with our findings, a recent study[55] has reported that deletion of a 241 kb interval within the gene desert (VS-250, mm10 chr3:66444310-66685547) is sufficient to deplete *Shox2* in the SV. This resulted in a hypoplastic SAN and abnormally developed venous valve primordia responsible for embryonic lethality[55]. We therefore concluded that loss of *Shox2* in hearts of GD$^{\Delta/\Delta}$ embryos results from inactivation of one (or more) SV/SAN enhancer(s) in the VS-250 interval (Fig. 4D). While our epigenomic analysis from whole hearts identified multiple elements with cardiac enhancer signatures (H3K27ac) (Fig. 1B), none was found to drive reproducible activity in embryonic hearts at E11.5 using transgenic reporter assays (Fig. 1C). To refine *Shox2*-associated cardiac enhancer predictions we performed ATAC-seq from mouse embryonic hearts at E11.5 and intersected the results with reprocessed open chromatin signatures from HCN4$^+$-GFP sorted SAN pacemaker cells of mouse hearts at P0, available from two recent studies[46,66] (Fig. 4D, Supplementary Data 3). Intersection of peak calls within the VS-250 interval identified multiple sites with overlapping accessible chromatin in embryonic hearts and perinatal SAN cells. While a subset of these candidate SAN enhancer elements overlapped DEs validated for non-cardiac activities (DE 3, 4, 7-12), the remaining open chromatin regions (+319, +325, +389, +405, +417, +515, +520) included yet uncharacterized elements showing variable enrichment for TBX5, GATA4 and/or TEAD TFs which are associated with SAN enhancer activation[46,67,68] (Fig. 4D, Supplementary Fig. 5A, Supplementary Data 3). To obtain complete functional validation coverage, we assayed these new putative SV/SAN enhancer elements by LacZ reporter transgenesis in mouse embryos (Fig. 4D, Supplementary Table 5). This analysis identified a single element located 325 kb downstream of *Shox2* (+325) that was able to drive reproducible LacZ reporter expression specifically in the cardiac SV region in a reproducible manner (Fig. 4E, Supplementary Fig. 5b) and showed interaction with the *Shox2* promoter in hearts at E11.5, as indicated by v4C analysis using viewpoints on the *Shox2* promoter and the +325 element itself (Fig. 4D, Supplementary Fig. 3B, 5C, Supplementary Data 2). To further define the core region responsible for the SV-specific activity we divided the 4kb-spanning +325 module into two elements: +325A and +325B (Fig. 4E, Supplementary Table 5). These elements overlapped in a conserved block of sequence (1.5 kb) that showed an open chromatin peak in embryonic hearts at E11.5 and SAN cells at P0, and also co-localized with TBX5 enrichment at E12.5

(Fig. 4E). Both +325A and +325B elements retained SV enhancer activity on their own in transgenic reporter assays, indicating that the core sequence is responsible for SV activity (Fig. 4E, Supplementary Fig. 5B). To identify cardiac TF interaction partners in enhancers at the motif level, we then established a general framework based on a former model of statistically significant matching motifs[69] and restricted to TFs expressed in the developing heart at E11.5 (Supplementary Data 4) (see Methods). This approach identified a bi-directional TBX5 motif in the active core [$P = 1.69e$-05 (+) and $P = 1.04e$-05 (-)] of the +325 SV enhancer module which, in addition to ChIP-seq binding, suggested direct recruitment of TBX5 (Fig. 4E, Supplementary Fig. 5D). In contrast, no motifs or binding of other established cardiac *Shox2* upstream regulators (e.g., Isl1) were identified in this core sequence (Supplementary Fig. 5A, 5D). In summary, our results identified the +325 module as a remote TBX5-interacting cardiac enhancer associated with transcriptional control of *Shox2* in the SV and thus likely required for SAN progenitor differentiation[36,55].

The mouse +325 SV enhancer core module is conserved in the human genome where it is located 268 kb downstream (+268) of the TSS of the *SHOX2* ortholog. Taking advantage of fetal left and right atrial (LA and RA) as well as left and right ventricular (LV and RV) tissue samples at post conception week 17 (pcw17; available from the Human Developmental Biology Resource at Newcastle University), we conducted H3K27ac ChIP-seq and RNA-seq to explore chamber-specific SV enhancer activity during pre-natal human heart development (Fig. 5A). These experiments uncovered an atrial-specific H3K27ac signature at the (+268) conserved enhancer module, matching the transcriptional specificity of *SHOX2* distinct from the ubiquitous profile of *RSRC1* in human hearts (Fig. 5A). This result indicating human-conserved activity prompted us to further investigate the developmental requirement of the SV enhancer in vivo. Therefore, we used CRISPR-Cas9 in mouse zygotes (CRISPR-EZ)[70] to delete a 4.4 kb region encompassing the +325 SV enhancer interval (SV-Enh$^{\Delta}$) (Fig. 5B, Supplementary Fig. 5E, F; Supplementary Tables 2, 3). F1 mice heterozygous for the SV enhancer deletion (SV-Enh$^{\Delta/+}$) were phenotypically normal and subsequently intercrossed to produce homozygous SV-Enh$^{\Delta/\Delta}$ embryos. ISH analysis indeed pointed to downregulation of *Shox2* transcripts in the SV region in SV-Enh$^{\Delta/\Delta}$ embryos at E10.5 and qPCR analysis at the same stage demonstrated a -60% reduction of *Shox2* in hearts of SV-Enh$^{\Delta/\Delta}$ embryos compared to WT controls (Fig. 5C). Despite this reduction of *Shox2* dosage in embryos, SV-Enh$^{\Delta/\Delta}$ mice were born at normal Mendelian frequency and showed no overt phenotypic abnormalities during adulthood. Together, these results imply that multiple gene desert enhancers participate in transcriptional control of *Shox2* in SAN progenitors, and that the +325 SV enhancer individually contributes as a core module to buffering of cardiac *Shox2* to protect from dosage-reducing mutations.

## A gene desert limb enhancer repertoire promotes stylopod morphogenesis

Due to the critical role of *Shox2* in proximal limb development we also addressed the functional requirement of the gene desert for skeletal

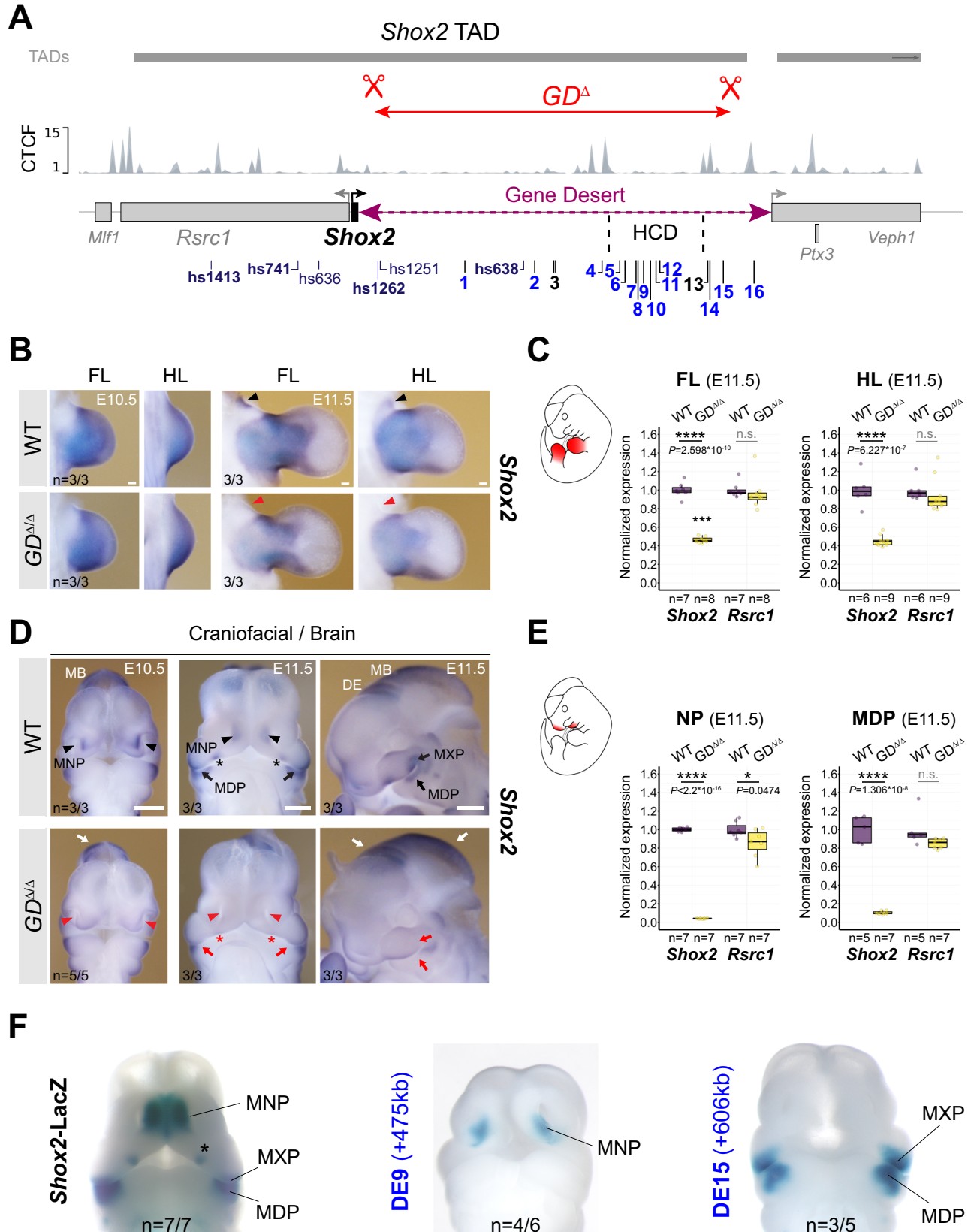

limb morphogenesis. *Shox2* is essential for stylopod formation and thus analysis of skeletal elements serves as an ideal readout for the study of enhancer-related *Shox2* dosage reduction in the proximal limb[21,29]. Neither knockout of the hs1262 proximal limb enhancer[21] nor the identification of new gene desert limb enhancers that all showed weak or restricted activities (DE4, DE6, DE10) (Fig. 1) was sufficient to

explain the -50% *Shox2* reduction observed in proximal fore- (FL) and hindlimbs (HL) of GD^Δ/Δ embryos (Fig. 3B, C). To refine our epigenomic limb enhancer predictions at the spatial level we reprocessed previously published ChIP-seq datasets from dissected proximal and distal limbs at E12[59] which revealed multiple proximal-specific H3K27ac peaks (Fig. 6A). These included several elements not significantly

**Fig. 3 | Gene desert deletion reduces *Shox2* in limb and craniofacial compartments. A** CRISPR/Cas9-mediated deletion of the intra-TAD *Shox2* gene desert interval (GD^Δ) (mm10, chr3:66365062-66947168). Vista (hs) and newly identified gene desert enhancers (1-16, active in blue) are displayed along with TAD interval and CTCF peaks from mESCs[64]. HCD, high-density contact domain (see Fig. 2). **B**, **D** ISH revealing spatial *Shox2* expression in fore- and hindlimb (FL/HL), craniofacial compartments, and brain in GD^Δ/Δ embryos compared to wildtype (WT) controls at E10.5 and E11.5. Red arrowheads and red arrows point to regions with severely downregulated or reduced *Shox2* expression, respectively. Red asterisk demarcates *Shox2* loss in the anterior portion of the palatal shelves. White arrows indicate regions (diencephalon, DE and midbrain, MB) without overt changes in *Shox2* expression. Scale bars, 500 μm (b) and 100 μm (d). **C**, **E** Quantitative mRNA analysis (qPCR) in limb and craniofacial tissues of WT and GD^Δ/Δ embryos. Box plots indicate interquartile range, median, maximum/minimum values (bars). Dots represent individual data points. ****$P < 0.0001$; *$P < 0.05$; n.s., not significant (two-tailed, unpaired *t*-test for qPCR). **F** DE9 and DE15 enhancer activities (Fig. 1C) overlap *Shox2* expression in medial nasal process (MNP) and maxillary-mandibular (MXP-MDP) regions, respectively, in mouse embryos at E11.5. Asterisk marks anterior palatal shelf. "n" indicates number of embryos per genotype, or transgene analyzed, with similar results. Source data are provided in the Source Data file.

enriched in H3K27ac maps from whole-mount limb tissue (Fig. 1B). Interestingly, multiple elements marked by H3K27ac in proximal limbs also showed H3K27me3 in distal limb mesenchyme reflecting compartment-specific bivalent epigenetic regulation[59]. With the goal to identify the complement of H3K27ac-marked elements that interact with the *Shox2* promoter we performed circular chromosome conformation capture (4C-seq) with a *Shox2* viewpoint from dissected proximal limbs at E12.5 (Fig. 6B, Supplementary Table 6). Processing of two replicates resulted in reproducible peaks which confirmed physical interaction between the *Shox2* promoter and each of the bona-fide proximal limb enhancers (PLEs) characterized previously: hs741 located in the upstream domain (U-dom) and hs1262 located in the gene desert (D-dom)[21,49] (Fig. 6A, B). Other prominent 4C-seq peaks in the gene desert co-localized with either previously validated enhancer elements with non-limb activities at E11.5 (DE1, 6, 9, 15) or non-validated elements with proximal limb-specific H3K27ac enrichment (+237 kb and +568 kb) (Fig. 6A, B). Epigenomic profiles further revealed that the *Shox2*-interacting DE4 (+407) element showing restricted LacZ activity in the proximal limb at E11.5 ($n = 2/5$) was unique in its H3K27ac pattern initiated past E10.5, while other (candidate) PLEs showed H3K27ac enrichment already present at E10.5 (Fig. 6A, Supplementary Fig. 6A). Therefore, we decided to analyze the spatiotemporal activities of newly identified (+237 kb, +568 kb) and seemingly temporally dynamic (DE4, +407) (candidate) limb enhancer regions using stable transgenic LacZ reporter mouse lines. For comparison, we also assessed the previously identified hs741 (termed PLE1) and hs1262 (PLE2) *Shox2* limb enhancers[21,49] (Fig. 6A, Supplementary Fig. 6A and Supplementary Table 7). Remarkably, at E12.5, each element on its own was able to drive reporter expression in the proximal fore- and hindlimb mesenchyme in a pattern overlapping *Shox2*, projecting a complement of at least five PLEs that contact *Shox2*, with four of those residing in the gene desert (PLE2-5) (Fig. 6A–C, Supplementary Fig. 6B). These activity patterns generally showed strong reporter signal in the peripheral mesenchyme of the stylopod and zeugopod elements (Fig. 6C, Supplementary Fig. 6B). *Shox2* expression is progressively downregulated within the differentiating chondrocytes of the proximal skeletal condensations of the limbs from E11.5, while its expression remains high in the surrounding mesenchyme and perichondrium[51,71–73]. In accordance, activities of the newly discovered elements (PLE3-5) remained excluded from the chondrogenic cores of the skeletal condensations, consistent with a role in shaping the *Shox2* expression pattern required for stylopodial chondrocyte maturation and subsequent osteogenesis[12,29]. PLE3 (+237) reporter activity was initiated in the proximal limb mesenchyme at E11.5 with persistent signal until E13.5 and most closely recapitulating the late *Shox2* expression pattern[29,51] (Supplementary Fig. 6B). Similarly, PLE4/DE4 (+407) activity emerged at E11.5 in the proximal-posterior (see also Fig. 1C) but extended in a more widespread fashion into distal limbs at later stages, in line with elevated H3K27ac in distal forelimbs at E12.5 (Fig. 6A, Supplementary Fig. 6A, B). PLE5 (+568) was initiated only at E12.5 and its activity remained restricted to the proximal-anterior (Supplementary Fig. 6B). Together, these diverse and partially overlapping enhancer activities pointed to dynamic interaction of *Shox2* gene desert enhancers during limb development. In addition, to achieve insight into PLE configuration at the chromatin level we performed 4C-seq with viewpoints at PLE2 and PLE4 which indicated the formation of a complex involving PLE1, 3 and 4, but not PLE2 (Supplementary Fig. 6C–E). These findings suggest that PLE interactions might not necessarily be restricted to U-dom or D-dom sub-compartments for *Shox2* regulation in the limb. Taken together, our results identify the gene desert as a multipartite *Shox2* limb enhancer unit with a potentially instructive role in the transcriptional control of stylopod morphogenesis.

Lastly, to evaluate the functional and phenotypic contribution of the gene desert to stylopod formation we combined our gene desert deletion allele with a *Prx1*-Cre conditional approach for *Shox2* inactivation[29,74]. This enabled limb-specific conditional deletion of *Shox2* on one allele (*Shox2^Δc*), paired with deletion of the gene desert on the other allele (GD^Δ), allowing to bypass embryonic lethality caused by the loss of cardiac *Shox2* (Fig. 7A, Supplementary Fig. 4). Remarkably, this abolishment of gene desert-mediated *Shox2* regulation in limbs led to a reduction of around 25–30% of *Shox2* transcripts in fore- and hindlimbs of GD^Δ/*Shox2^Δc* embryos at E11.5, as compared to *Shox2^Δc/+* heterozygote controls (Fig. 7B). This reduction surpassed the reported effect of PLE1(hs741);PLE2(hs1262) double enhancer loss in *Shox2*-deficient background in hindlimbs which was predominantly associated to PLE1 (~15% reduction), an enhancer located outside of the gene desert[21,29]. As expected, endogenous PLE2 removal via the LHB^Δ allele in limb-specific *Shox2* sensitized background failed to result in significant *Shox2* reduction in embryonic forelimbs of LHB^Δ/*Shox2^Δc* embryos compared to *Shox2^Δc/+* controls (Supplementary Figs. 7, 8A), suggesting relevant limb-specific functional contributions of PLEs other than PLE2/hs1262 within the gene desert. In agreement, at perinatal stage GD^Δ/*Shox2^Δc* mutants showed more severe shortening of the stylopod than PLE1(hs741);PLE2(hs1262) double enhancer knockouts in *Shox2*-sensitized conditions[21,29], with an approximate 60% reduction in humerus length and 80% decrease in femur extension in GD^Δ/*Shox2^Δc* newborn mice (Supplementary Fig. 8B, C). In addition, micro-computed tomography (μCT) from respective adult mouse limbs at P42 showed significant humerus length reduction of approximately 40% and decreased femur length of about 50% (Fig. 7C, D). Our results thus demonstrate an essential role of the gene desert in proximal limb morphogenesis and imply a significant functional contribution of the PLE2-5 modules to spatiotemporal control of *Shox2* dosage in the limb.

In summary, our study identifies the *Shox2* gene desert as an essential and dynamic chromatin unit encoding an array of distributed tissue-specific enhancers that coordinately regulate stylopod formation, craniofacial patterning, and SAN pacemaker dependent embryonic progression (Fig. 8A–C). The arrangement of the enhancers appears modular but distributed in terms of tissue-specificities (Fig. 8A). While craniofacial and neuronal gene desert enhancers are hallmarked by driving mostly distinct subregional activities, limb enhancers (PLEs) show more overlapping activity domains, pointing to potential redundant intra-gene desert enhancer interactions. Hereby, the detection of a high-density contact domain (HCD) suggests that sub-TAD compartmentalization could further contribute to modulation of subregional enhancer activities (Fig. 8B). Finally, the

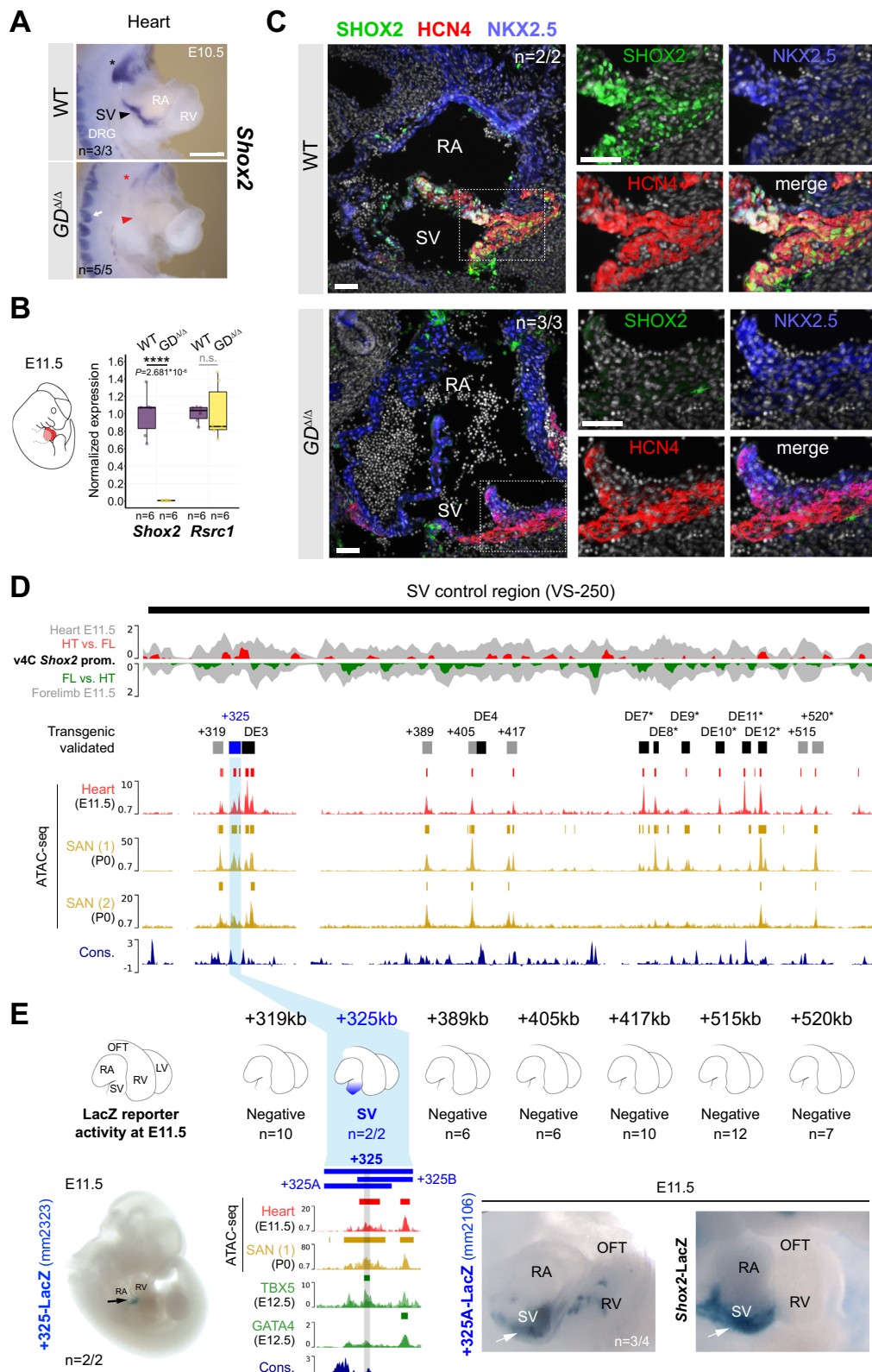

demonstrated phenotypic requirement of the *Shox2* gene desert for multiple developmental processes underscores the importance of functional studies focused on the non-coding genome for better mechanistic understanding of congenital abnormalities (Fig. 8C).

## Discussion
There is now evidence that dismantling of duplicates of ancient genomic regulatory blocks (GRBs) led to the emergence of gene deserts enriched in the neighborhood of regulatory genes such as TFs[75]. Functional assessment of TF gene deserts, including those in the *Hoxd* and *Sox9* loci, revealed that distal long-range enhancers represent critical *cis*-regulatory modules that control subregional expression domains through interaction with target gene promoters in a spatiotemporal manner[6,7,76,77]. Gene deserts can thus be conceived as genomic units coordinating dynamic enhancer activities in specific developmental processes, such as *HoxD*-dependent digit formation,

**Fig. 4 | Gene desert-mediated transcriptional control of cardiac *Shox2* is essential for embryonic viability. A** ISH revealing *Shox2* downregulation in the cardiac sinus venosus (SV) (red arrowhead) and nodose ganglion of the vagus nerve (red asterisk) in GD$^{\Delta/\Delta}$ embryos at E10.5. White arrow indicates normal *Shox2* expression in the dorsal root ganglia (DRG) of GD$^{\Delta/\Delta}$ embryos. Scale bar, 100 μm. **B** Quantitative PCR (qPCR) revealing depletion of *Shox2* in GD$^{\Delta/\Delta}$ hearts compared to WT controls at E11.5. Box plots indicate interquartile range, median, maximum/minimum values (bars). Dots represent individual data points. ****$P < 0.0001$; n.s., not significant (two-tailed, unpaired *t*-test). **C** Co-localization of SHOX2 (green), HCN4 (red) and NKX2-5 (blue) in hearts of GD$^{\Delta/\Delta}$ and WT control embryos at E11.5. SHOX2 is lost in the HCN4-marked SAN pacemaker myocardium in absence of the gene desert (dashed outline). Nuclei are shown in gray. Scale bars, 50 μm. **D** SAN enhancer candidate regions in the gene desert interval (VS-250) essential for *Shox2* expression in the SV[55]. Top: Virtual 4 C (v4C) *Shox2* promoter interaction signatures in embryonic hearts (HT) and limbs (FL) (gray) at E11.5 overlapped with HT (red)

and FL (green) -specific subtraction profiles. Below: ATAC-seq tracks from embryonic hearts at E11.5 and SAN cells from sorted *Hcn4*-GFP mouse hearts at P0 (Supplementary Data 3)[46,66]. Desert enhancers (DEs) (black) and putative SAN enhancer elements with distance to the *Shox2* TSS in kb (+) are indicated. Cons, vertebrate conservation track by PhyloP. **E** Transgenic LacZ reporter validation in mouse embryos at E11.5. Left: the +325 element drives transgenic LacZ reporter expression exclusively in the SV. Right: the +325A subregion drives *Shox2*-overlapping SV activity, similar to +325B (Supplementary Fig. 5). The interval shared between +325A/B subregions contains a conserved core element (marked gray) that interacts with TBX5 in embryonic hearts at E12.5[67]. "n" denotes fraction of biological replicates with reproducible results. Single numbers represent the total of transgenic embryos obtained, including those without staining. RA, right atrium. RV, right ventricle. OFT, outflow tract. Corresponding Vista Enhancer IDs (mm) are listed in Supplementary Table 5. Source data are provided in the Source Data file.

and can be also hi-jacked by evolutionary processes to enable phenotypic diversification[9,12,78]. Silencer modules, insulating TAD boundaries and tethering elements (promoting long-range interactions) are involved in restriction or modulation of E-P interactions in metazoan genomes and can further contribute to gene desert functionality[79–81]. Recent studies also indicated that functional RNAs, such as lncRNAs or circRNAs, represent elements with enhancer-modifying or distinct regulatory potential within gene deserts[82]. Importantly, human disease-associated nucleotide variants in gene deserts are frequently linked to enhancer function, contributing to the spectrum of enhanceropathies[83–85]. Furthermore, deletions, inversions and duplications can alter or re-distribute interaction of gene desert enhancers with target gene promoters leading to congenital malformation or syndromes[10,19,20]. Despite these critical implications, the enhancer landscapes and related chromatin topology of most gene deserts near developmental genes remain incompletely characterized at the functional level[86]. In the current study, we addressed the functional necessity and *cis*-regulatory architecture of a gene desert flanking the *Shox2* transcriptional regulator, a critical determinant of embryogenesis and essential for limb, craniofacial and SAN pacemaker morphogenesis[39,49,87]. We identify the *Shox2* gene desert as a reservoir for highly subregional, tissue-specific enhancers underlying pleiotropic *Shox2* dosage by demonstrating essential contributions to stylopod morphogenesis, craniofacial patterning, and SV/SAN development. Our findings support a model in which gene deserts provide a scaffold for preferential chromatin domains that generate enhancer-mediated cell type or tissue-specific *cis*-regulatory output based on the integration of upstream signals.

Interpretation of gene desert function is dependent on accurate functional prediction of enhancer activities embedded in the genomic interval. Our approach using ChromHMM-filtered H3K27ac signatures from bulk tissues across a large range of embryonic stages (derived from ENCODE) serves as a baseline for the mapping of tissue-specific enhancer activities. However, while H3K27ac is known as the most specific canonical mark for active enhancers, it appears to not include all enhancers[88–90]. For example, recent studies evaluating H3K27ac-based tissue-specific enhancer predictions in mouse embryos revealed a substantial number of false-positives[57,91]. In turn, a significant fraction (~14%) of validated in vivo enhancers were lacking enrichment of any canonical enhancer marks (ATAC-seq, H3K4me1, H3K27ac)[91]. In line with these observations, our transgenic reporter validation in many cases revealed more restricted or even distinct in vivo enhancer activities than those predicted by epigenomic marks. Such discrepancies might be partially originating from the use of bulk tissues or limited sensitivity of profiling techniques. In accordance, refinement of enhancer predictions using region-specific open chromatin data in combination with chromatin conformation capture (C-HiC, 4C-seq) enabled us to identify critical subregional cardiac and proximal

limb enhancers missed by the initial and rather stringent epigenomic prediction approach.

Genomic deletion analysis uncovered an important functional role of the gene desert for pleiotropic expression and progression of embryonic development, the latter through direct control of *Shox2* in SAN pacemaker progenitors. Consistent with our findings, a parallel study narrowed the region essentially required for cardiac *Shox2* expression to a 241 kb gene desert interval (termed VS-250)[55]. Here, we have identified a human-conserved SV enhancer (+ 325) located within this essential interval and specifically active in the SV/SAN region to maintain robust cardiac *Shox2* levels. These results add to recent progress in uncovering SAN enhancers of cardiac pacemaker regulators, including also *Isl1* or *Tbx3*[46,55]. Such findings not only shed light on the wiring of the GRNs driving mammalian conduction system development but also offer the opportunity to identify mutational targets linked to defects in the pacemaker system, such as arrhythmias[65]. Interestingly, removal of the *Isl1* SAN enhancer (ISE) in mice, as for our +325 *Shox2* enhancer, led to reduced target gene dosage but without subsequent embryonic or perinatal lethality[46]. These instances indicate that the GRNs orchestrating SAN pacemaker development are buffered at the *cis*-regulatory level, which can be enabled through partially redundant enhancer landscapes[21,92]. Similar to the binding profile of the +325 *Shox2* SV enhancer, a TF network involving GATA4, TBX5 and TEAD has been implicated in ISE activation, confirming a key role of TBX5 in the activation of SAN enhancers in working atrial myocardium, while pacemaker-restricted identity may be established by repressive mechanisms[45,46,65]. ISE activity was also correlated with abnormal SAN function in adult mice and found to co-localize with resting heart rate SNPs, indicating potentially more sensitive GRN architecture in humans[46]. Intriguingly, coding and non-coding variants in the human *SHOX2* locus were recently associated with SAN dysfunction and atrial fibrillation, underscoring the value of human-conserved SAN enhancer characterization for functional disease variant screening[40,42,93,94].

Arrangements of distributed enhancer landscapes conferring robust and cell type-specific transcription emerged as a common feature of metazoan gene regulatory architecture[95–97]. Gene deserts may thus not only function to promote robust expression boundaries and/or phenotypic resilience, but also represent a platform enabling evolutionary plasticity[9,75]. The conventional model of enhancer additivity based on individual small and stable regulatory contributions is likely predominant in gene deserts[98]. In support, we uncovered at least four *Shox2*-associated gene desert enhancers (PLE2-5) with overlapping activities in the proximal limb mesenchyme. Such regulatory architecture resembles the multipartite enhancer landscapes in *Indian Hedgehog* (*Ihh*) or *Gremlin1* loci, which as *Shox2* are involved in spatiotemporal coordination of proximal-distal limb identities with chondrogenic cues[24,26]. Our study further reveals gene desert

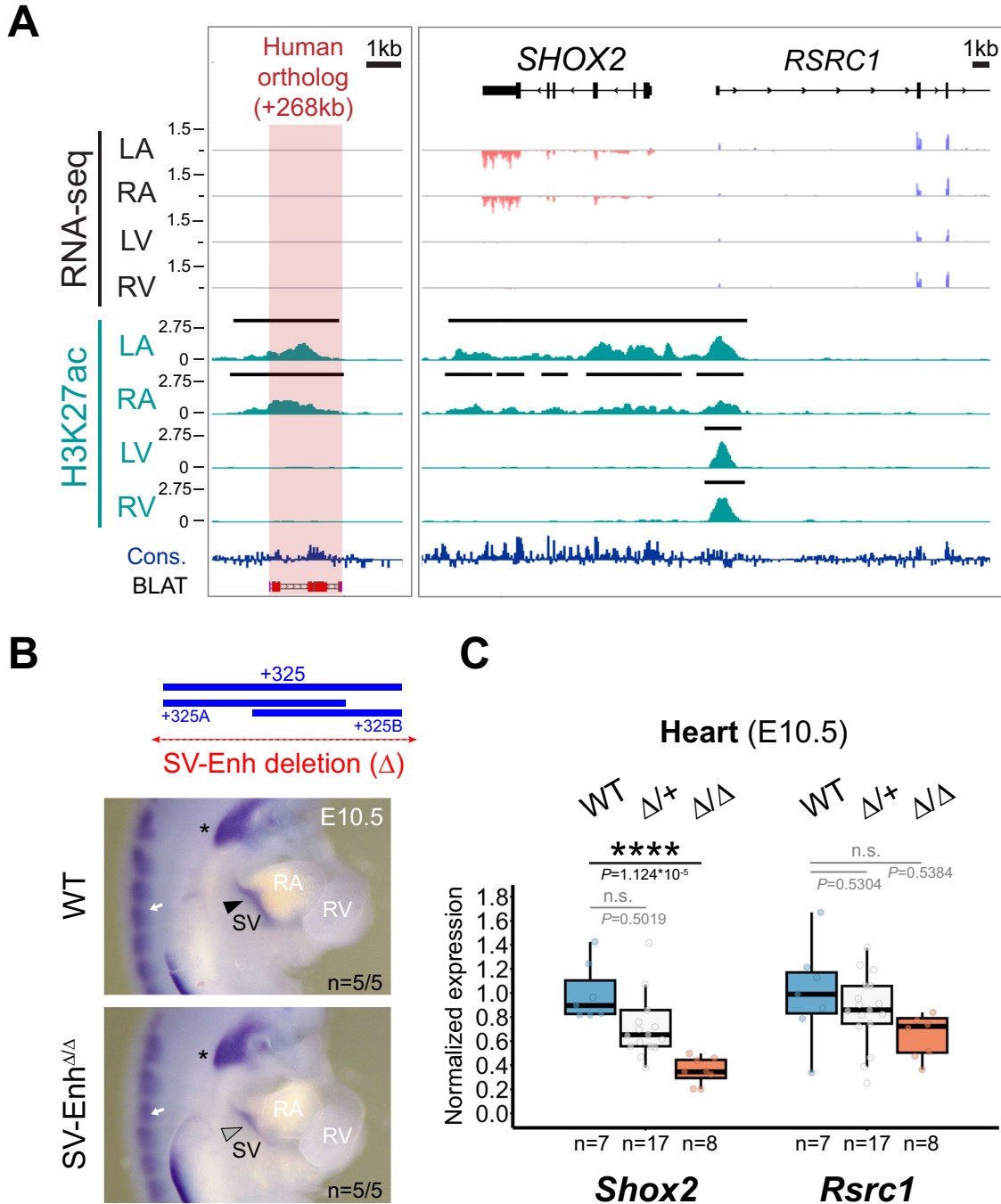

**Fig. 5 | Enhancer-mediated transcriptional robustness safeguards *Shox2* in the heart. A** H3K27 acetylation ChIP-seq (H3K27ac) and RNA-seq profiles from human fetal heart compartments at post conception week 17 (pcw17) across the human orthologous sequence of the +325-mouse sinus venosus (SV) enhancer and the *SHOX2* interval. The left ventricle (LV) dataset has been previously published[115]. +268, distance to *SHOX2* TSS. Cons, mammalian conservation by PhyloP. **B** Top: Generation of a + 325 SV enhancer deletion (4.4 kb) allele in mice (SV-Enh^Δ). Below: *Shox2* mRNA distribution (ISH) in SV-Enh^{Δ/Δ} compared to WT mouse embryos at E10.5. Arrowhead points to downregulated *Shox2* in the SV. Asterisk and arrow mark *Shox2* expression in the nodose ganglion of the vagus nerve and dorsal root ganglia (DRG), respectively. **C** qPCR analysis of *Shox2* and *Rsrc1* mRNA levels in SV-Enh^{Δ/+} and SV-Enh^{Δ/Δ} embryonic hearts at E10.5 compared to WT controls. Box plot indicates interquartile range, median, maximum/minimum values (bars) and individual biological replicates (*n*). *P*-values are shown, with ****P < 0.0001 (two-tailed, unpaired *t*-test). Three outliers, two datapoints of *Shox2* Δ/+ replicates and one for *Rsrc1* (Δ/Δ), are outside of the scale shown. N.s., not significant. "n" indicates number of biological replicates analyzed, with similar results. LA, left atrium. RA, right atrium. RV, right ventricle. Source data are provided in the Source Data file.

enhancers with seemingly unique tissue specificities, such as the craniofacial DE9 and DE15 elements driving *Shox2*-overlapping reporter expression in the nasal process and maxillary-mandibular region, respectively. DE15 may be involved in jaw formation as *Shox2* inactivation in cranial neural crest cells in the maxilla-mandibular junction leads to dysplasia and ankylosis of the TMJ in mice[30].

Our C-HiC experiments indicated that the repertoire of *Shox2* interacting elements (e.g., enhancers) is confined to the overarching TAD, without apparent cross-TAD boundary interactions[99]. The observed U-dom and D-dom assemblies (as evidenced by loop anchors) might reflect dynamic loop structures to facilitate *Shox2* promoter scanning similar to the organization at *HoxA* and *HoxD* loci

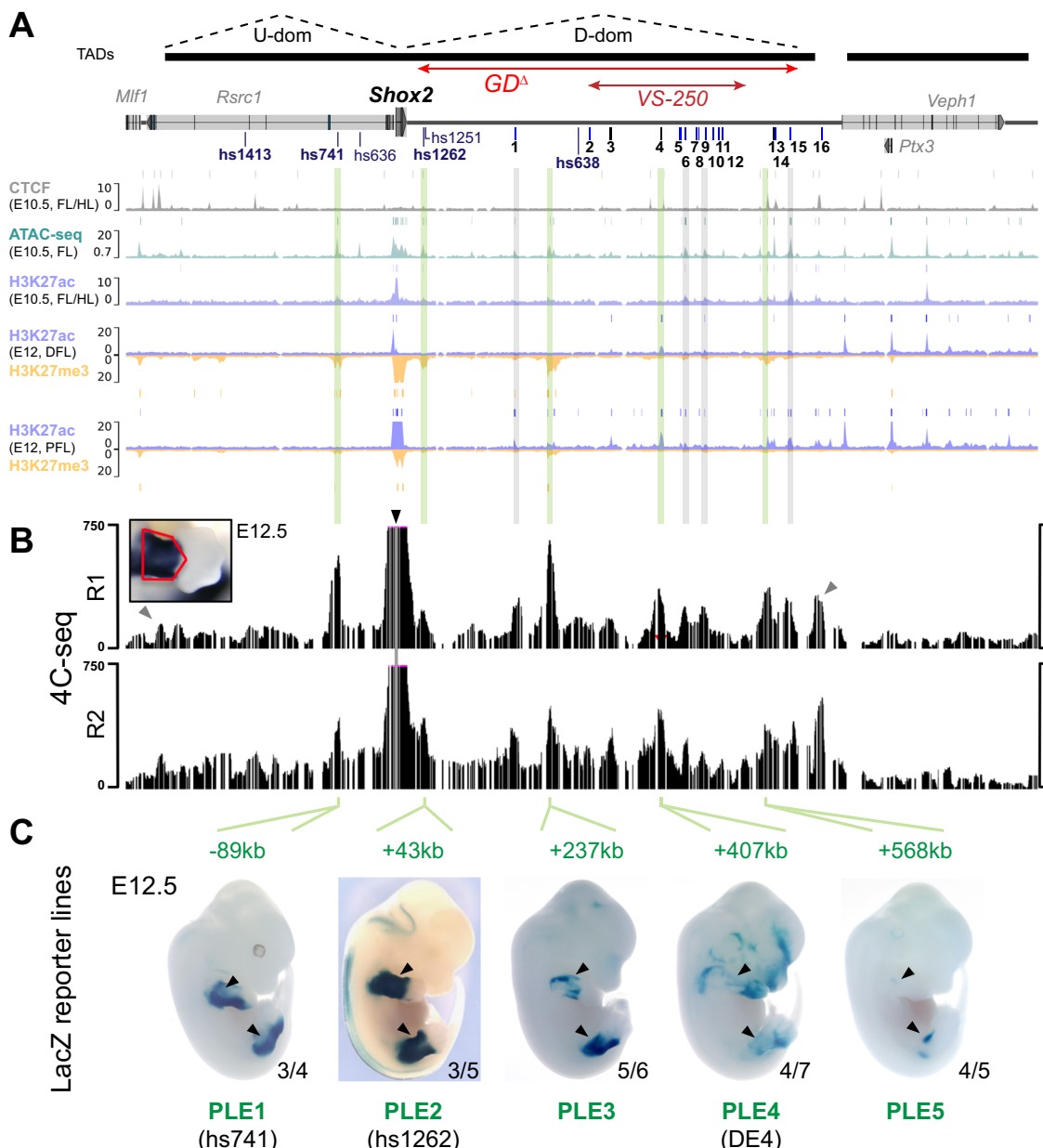

**Fig. 6 | The gene desert encodes a series of distributed proximal limb enhancers (PLEs) with subregional specificities. A** Re-processed ChIP-seq datasets from mouse embryonic limbs at E10.5 (CTCF, ATAC-seq, H3K27ac) and E12 (H3K27ac, H3K27me3) showing epigenomic profiles at the *Shox2* locus[59,63,144]. Bars above each track represent peak calls across replicates. DFL, distal forelimb. PFL, proximal forelimb. On top: TAD extension in mESCs[64] (black bars) with desert enhancers identified in Fig. 1 (DEs 1-16; blue indicates validated activity). The extension of the gene desert deletion (GD^Δ) and SV control region (VS-250) is indicated. **B** 4C-seq interaction profiles from two independent biological replicates (R1, R2) of proximal forelimbs at E12.5 (red outline). Black arrowhead indicates the 4C-seq viewpoint at the *Shox2* promoter. Gray arrowheads point to CTCF-boundaries of the *Shox2*-TAD. Green lines (in A) indicate *Shox2*-interacting elements with putative proximal limb activities. Gray lines mark 4C-seq peaks overlapping previously validated DEs without such activities (Fig. 1C). **C** Identification of proximal limb enhancers (PLEs) through transgenic LacZ reporter assays in mouse embryos at E12.5. Embryos shown are representatives from stable transgenic LacZ reporter lines (Supplementary Fig. 6A). Reproducibility numbers from original transgenic founders are listed for each element (bottom right).

that promotes nested and collinear gene expression[7,100]. C-HiC analysis also uncovered a high-density contact domain (HCD) emerging only in heart tissue. The absence of convergent CTCF sites flanking the HCD might reflect that a subset of contact domains form independently of cohesin-mediated loop extrusion, for example based on self-aggregation of regions carrying identical epigenetic marks or the emergence of globule structures resulting from phase separation[101–104]. Interestingly, the HCD genomic interval harbors several validated enhancers that were inactive in the embryonic heart (DE5-12). An intriguing hypothesis raised by these observations is that HCDs could act to topologically sequester regulatory regions for modulation of

target gene interaction in a tissue-specific manner. While such domains might have inhibiting or augmenting impact on tissue-specific regulatory interactions, a neutral effect may also be possible.

From a disease perspective, our findings also expand on former analyses demonstrating that *Shox2* gene desert limb and hindbrain enhancer activities emerge within the similar-sized gene desert flanking the human *SHOX*[49,105]. Pointing to functional homology with the mouse *Shox2* regulatory region, disruption of enhancers within the gene desert downstream of *SHOX* has been associated with Léri-Weill dyschondrosteosis (LWD) and idiopathic short stature (ISS) syndromes in a significant fraction of cases[106]. Furthermore, *SHOX* haploinsufficiency is

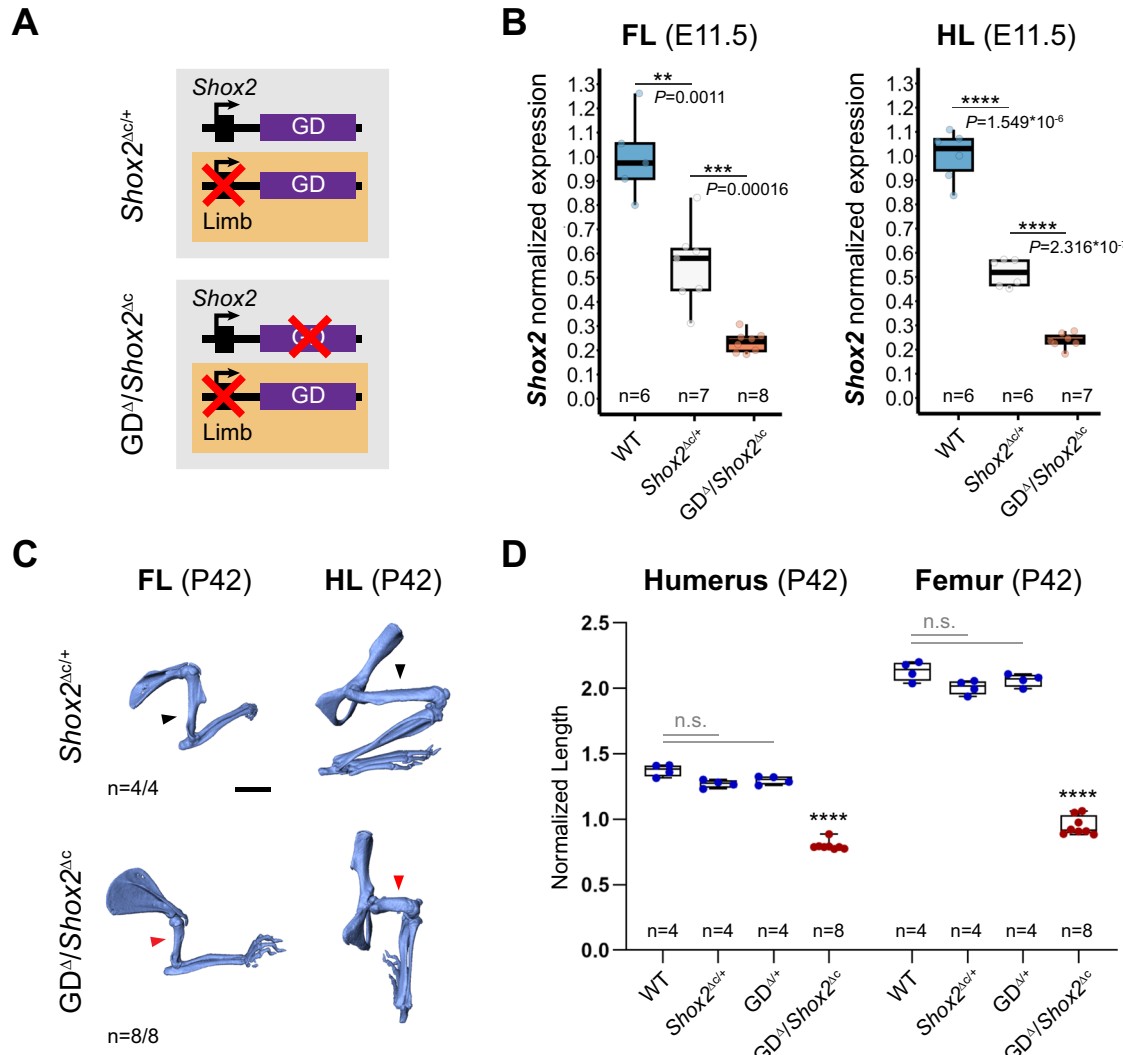

**Fig. 7 | Limb-specific loss of gene desert function leads to *Shox2* down-regulation and defective stylopod morphogenesis. A** Schematics illustrating gene desert inactivation (GD$^\Delta$) in the presence of reduced limb *Shox2* dosage based on *Prx1*-Cre-mediated *Shox2* deletion (*Shox2*$^{\Delta c}$). **B** *Shox2* transcript levels determined by qPCR in fore- (FL) and hindlimbs (HL) of wildtype (WT), *Shox2*$^{\Delta c/+}$ and GD$^\Delta$/*Shox2*$^{\Delta c}$ embryos at E11.5. One outlier (FL WT datapoint) is outside of the scale shown. **C** Micro-CT scans of fore (FL)- and hindlimb (HL) skeletons of GD$^\Delta$/*Shox2*$^{\Delta c}$ and *Shox2*$^{\Delta c/+}$ control mice at postnatal day 42 (P42). Red arrowheads point to severely reduced stylopods in GD$^\Delta$/*Shox2*$^{\Delta c}$ individuals compared to controls (black

arrowheads). "n", number of biological replicates with reproducible results. Scale bar, 5 mm. All images at same scale. **D** Micro-CT stylopod quantification at P42 reveals significant reduction of stylopod (humerus/femur) length in GD$^\Delta$/*Shox2*$^{\Delta c}$ mice compared to WT ($P = 9.6 \times 10^{-14}/P = 9.6 \times 10^{-14}$), *Shox2*$^{\Delta c/+}$ ($P = 1.27 \times 10^{-13}/P = 9.6 \times 10^{-14}$) and GD$^{\Delta/+}$ ($P = 1.27*10^{-13}/P = 9.6 \times 10^{-14}$) controls. Box plot indicates interquartile range, median, maximum/minimum values (bars) with dots representing individual biological replicates (n). ****$P < 0.0001$; ***$P < 0.001$; **$P < 0.01$; n.s., non-significant (two-tailed, unpaired *t*-test for qPCR; ANOVA for micro-CT). Source data are provided in the Source Data file.

directly associated with the skeletal abnormalities observed in LWD and Turner syndrome, the latter also involving craniofacial abnormalities[107–109]. One study has also found a link between neuro-developmental disorders and microduplications at the *SHOX* locus, suggesting that such perturbations may alter neural development or function[110]. Thus, considering the overlapping expression patterns and critical functions of human *SHOX* and mouse *Shox2*, our results provide a blueprint for the investigation of *SHOX* regulation in the hindbrain, thalamus, pharyngeal arches, and limbs[111,112]. It will be particularly interesting to determine whether "orthologous" craniofacial, neural and/or limb enhancers exist, and whether human *SHOX* enhancers share motif content or other enhancer grammar characteristics with mouse *Shox2* enhancers. Indeed, in a recent example orthologous enhancer-like sequence was identified 160 kb downstream of human *SHOX* and 47 kb downstream of mouse *Shox2*, respectively, and drove overlapping activities in the hindbrain[49,105]. Such enhancers presumably originate from a single ancestral *Shox* locus, preceding the duplication

of *Shox* and *Shox2* paralogs and are therefore considered evolutionary ancient. Within this context, future comparative studies should include a search for deeply conserved orthologs of *SHOX* and *SHOX2* enhancers in basal chordates such as amphioxus, which express their single *Shox2* gene in the developing hindbrain[113]. The recent identification of orthologous *Islet* gene enhancers in sponges and vertebrates demonstrate the promise of such an approach[114]. Taken together, functional enhancer characterization along with refined enhancer grammar and 3D interactions at the cell type level will likely be key to resolve the regulatory complexity inherent to distributed enhancer landscapes and to understand how transcriptional dynamics and morphological complexity are rooted in gene deserts.

## Methods
### Ethics statement
This research complies with all relevant ethical regulations. All aspects of this study involving human tissue samples were reviewed and

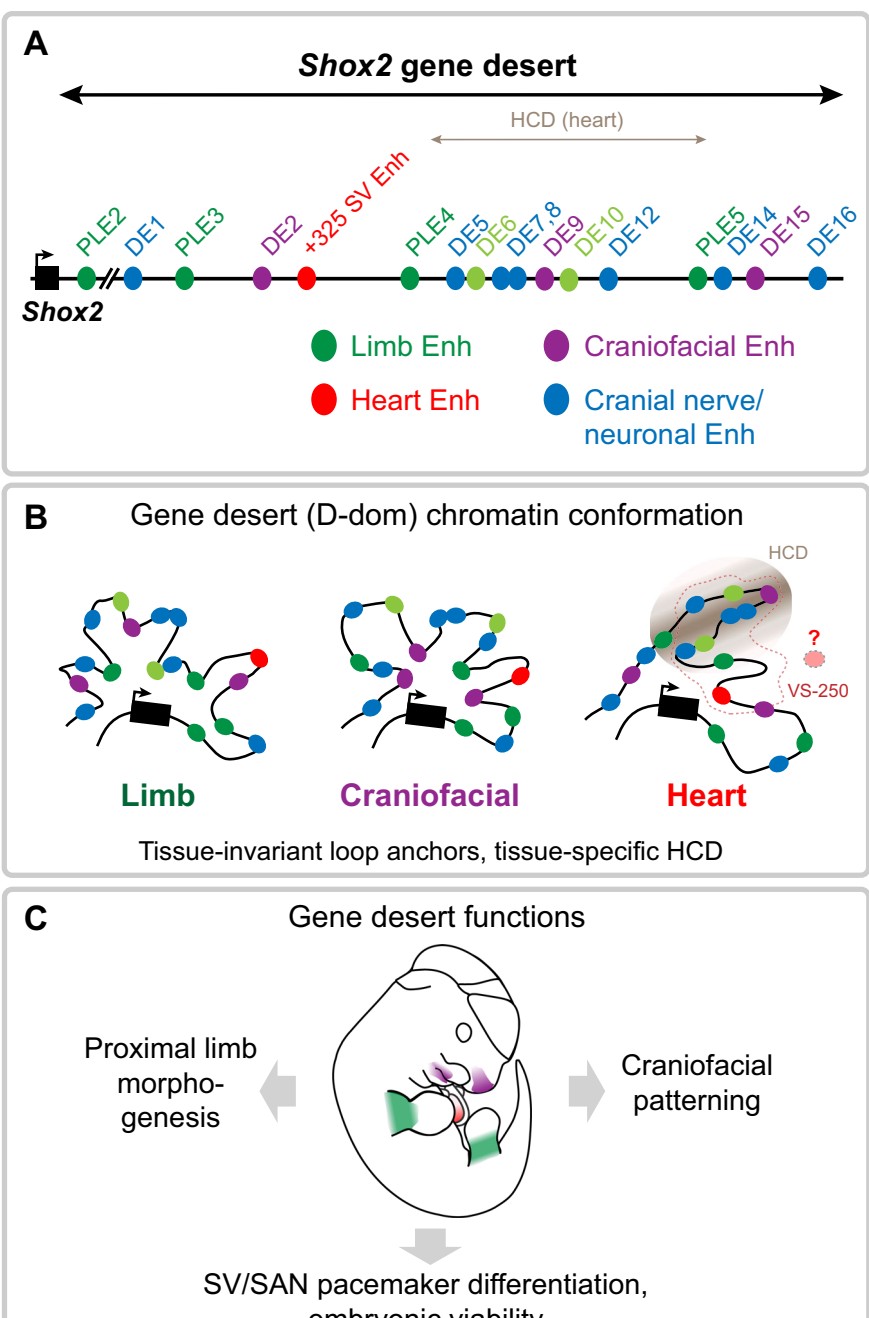

**Fig. 8 | Graphical summary. A** Identification of the *Shox2*-flanking gene desert as a reservoir for distributed transcriptional enhancers with *Shox2*-overlapping activities in limb, craniofacial, cardiac, and cranial nerve/neuronal cell populations. Light green indicates limb enhancers with highly subregional or limb type restricted activities. **B** The *Shox2* gene desert encodes distributed tissue-specific enhancers that are englobed in a dynamic chromatin domain (D-dom) with tissue-invariant loop anchors and a cardiac-specific high-density contact domain (HCD) that may influence the activity of contained enhancers. Additional gene desert enhancers are likely to participate in the regulation of cardiac *Shox2* in SAN progenitors. Dashed line demarcates the VS-250 interval essential for cardiac *Shox2* expression[55]. **C** Cumulative functions of gene desert enhancers orchestrate pleiotropic *Shox2* expression essential for proximal limb morphogenesis, craniofacial patterning, and cardiac pacemaker development.

approved by the Human Subjects Committee at Lawrence Berkeley National Laboratory (LBNL) under Protocol Nos. 00023126 and 00022756. All animal work at Lawrence Berkeley National Laboratory (LBNL, CA, USA) was reviewed and approved by the LBNL Animal Welfare Committee under protocol numbers 290003 and 290008. All animal work at the University of Calgary was reviewed and approved by the Life and Environmental Sciences Animal Care Committee (LESACC) under protocols AC21-0005 and AC21-0006, and in accordance with Canadian Council on Animal Care guidelines as approved by the University of Calgary (protocol AC13-0053). All animal work in Switzerland was reviewed and approved by the regional commission on Animal Experimentation and the Cantonal Veterinary Office of the city of Bern (protocol BE96/20).

Fetal human heart samples were obtained from the Human Developmental Biology Resource's Newcastle site (HDBR, hdbr.org), in compliance with applicable state and federal laws. The National Research Ethics Service reviewed the HDBR study under REC Ref 23/NE/0135, and IRAS project ID: 330783 in compliance with requirements

from the National Health Services for research within the UK and overseas. HDBR is a non-commercial entity funded by the Wellcome Trust and Medical Research Council. Fetal tissue donation is confidential, anonymized, completely voluntary with fully informed and explicitly documented written consent, and the participants do not receive compensation. In accordance, no identifying information for human samples in this study was shared by HDBR. More information about HDBR policies and ethical approvals can be accessed at https://www.hdbr.org/ethical-approvals.

### Human samples

Primary ChIP-seq and RNA-seq data from human cardiac compartments of a single embryo (sex: XY) at post conception week (PCW) 17 were generated de novo in this study (RV, LA, RA) or retrieved from our previous study (LV[115]) for analysis. Biopsies from cardiac compartments were collected at HDBR, and all embryonic samples were shipped on dry ice and stored at −80 °C until processed, as previously described[115,116]. ChIP-seq and RNA-seq data of cardiac compartments at PCW17 are presented in this study.

### Animal studies and experimental design

Mice used for transient transgenic reporter analysis (see section below) and mice of the GD$^\Delta$ line (strain: FVB/NJ) were housed at the LBNL Animal Care Facility, which is fully accredited by AAALAC International. Stable transgenic reporter mouse lines (strain: CD-1; see section below) and mice of the GD$^\Delta$ (strain: mixed FVB/C57BL/6NCrl) and LHB$^\Delta$ (strain: C57BL/6NCrl) genomic deletion lines were housed at the Life and Environmental Sciences Animal Resource Centre at the University of Calgary accredited by the Canadian Council on Animal Care. Mice of the SV-Enh$^\Delta$ line (strain: FVB/NRj) were housed at the Central Animal Facilities (CAF) of the Experimental Animal Center, University of Bern. The CAF runs upon approval of the Cantonal Authority, with husbandry license BE02/2022.

All mice were maintained with water supply on a 12:12 light-dark cycle, with relative humidity set at 30–70% (LBNL, University of Bern) or 20–50% (University of Calgary), and a temperature of 20–26.2 °C (LBNL, Calgary) or 22 °C +/− 2 °C (Bern). Mice at LBNL were housed in standard micro-isolator cages on hard-wood bedding with enrichment consisting of crinkle-cut naturalistic paper strands and fed on ad libitum PicoLab Rodent Diet 20 (5053). Mice at the University of Calgary were house in Tecniplast Blue Line IVC cages on hard-wood aspen chip bedding (autoclaved) with enrichment consisting of crinkle-cut naturalistic paper strands and a Cocoon nestlet (5800), while maintained on ad libitum irradiated PicoLab Mouse Diet 20 (5058). Mice at the University of Bern were housed in standard IVC cages GM500 Tecniplast, on Safe® Aspen wood granulate bedding with enrichment consisting of Pura® crinkle brown kraft paper nesting material, Pura® Brick Aspen Chew Block, and red polycarbonate mouse house and fed on Kliba Nafag standard breeding (3800) and maintenance diet (3430). All mice were health checked and monitored daily for food and water intake by trained personnel. Euthanasia at LBNL and University of Bern was performed in the home cage using $CO_2$ asphyxiation while ensuring gradual fill and displacement rate. Euthanasia at University of Calgary was performed by cervical dislocation after loss of consciousness induced by isoflurane anesthesia administered by the bell-jar method (250 μl on a gauze in a one liter container).

Animals of both sexes were used in these analyses. Sex was not considered as a variable in our embryonic studies since limb, craniofacial or heart development are expected to show minimal differences at the respective early stages of development. Skeletal analysis at P0 included both sexes as we did not expect normalized skeletal growth at P0 to show significant sex-based differences. Sex was tracked in mice used for micro-CT measurements at P42 and no significant sex-specific differences were observed after normalization. Unless specified otherwise, mice between 6 to 30 (predominantly 6 to 10) weeks of age

were used for breeding to generate embryos, newborns or adults analyzed in this study. Sample size selection strategies were conducted as follows:

**Transgenic mouse assays.** Sample size selection and scoring criteria were based on our experience of performing transgenic mouse assays for >3000 total putative enhancers (VISTA Enhancer Browser: http://enhancer.lbl.gov). Mouse embryos were excluded from further analysis if they did not encode the reporter transgene or if the developmental stage was not correct. Transgenic results were confirmed in at least three (for Hsp68-LacZ or βlacZ random integration) or two (for enSERT) independent biological replicates, based on criteria consistent with the pipeline established for the VISTA Enhancer Browser[117].

**Knockout mice.** Sample sizes were selected empirically based on our previous studies[21,22] and the minimal number of biological replicates analyzed per experiment is mentioned in the respective experimental sections below. Newborn mice at P0 (alizarin red/alcian blue staining) and adult mice at P42 (micro-CT) were used to assess limb skeletal phenotypes. All phenotypic characterization of knockout embryos and mice employed a matched littermate selection strategy. Embryonic littermates and samples from genetically modified animals were dissected and processed blind to genotype. Skeletons at P0 and P42 were measured randomized and blinded to genotype.

### In vivo transgenic reporter analysis

Transgenic reporter analysis of all elements except PLEs were performed in FVB mouse embryos (strain: FVB/NJ) at LBNL, as previously described[117]. Predicted enhancer elements were PCR-amplified from mouse genomic DNA (Clontech) and cloned into a Hsp68-LacZ expression vector (Addgene #170102) for random integration using Gibson assembly. For higher accuracy in absence of position effects, the +325 SV enhancer element was analyzed in an analogous manner but using a LacZ construct with a β-globin minimal promoter (Addgene #227000) for site-directed integration at the neutral H11 locus (enSERT)[27,118]. The sequence of the cloned constructs was confirmed via Sanger sequencing. Transgenic mice were generated via pronuclear injection[117]. Briefly, Hsp68-LacZ constructs were diluted in microinjection (MI) buffer (10 mM Tris, pH 7.5; 0.1 mM EDTA) and injected at 1.5 ng/μL for random integration. For enSERT, sgRNAs (50 ng/μl) targeting the H11 locus and Cas9 protein (Integrated DNA Technologies catalog no. 1081058; final concentration: 20 ng/μl) were mixed in microinjection buffer (10 mM Tris, pH 7.5; 0.1 mM EDTA). Mixes were injected into pronuclei of single-cell stage fertilized FVB/NJ (Jackson Laboratory; Strain#:001800) embryos obtained from the oviducts of super-ovulated 7–8 weeks old FVB/NJ females mated to 7–8 weeks old FVB/ NJ males. The injected embryos were cultured in M16 medium supplemented with amino acids at 37 °C under 5% $CO_2$ for ~2 h and transferred into the uteri of pseudo-pregnant CD-1 (Charles River Laboratories; Strain Code: 022) surrogate mothers. Embryos were collected for Beta-galactosidase staining experiments at embryonic days 10.5 or 11.5, as described[117]. Briefly, embryos were fixed with 4% paraformaldehyde (PFA) for 15 or 20 min (for stages E10.5 and E11.5, respectively) and stained overnight in X-gal stain while rolling at room temperature. The embryos were genotyped for the presence of the transgenic construct. Embryos positive for transgene integration and at the correct developmental stage were used for analysis and imaged on a Leica MZ16 microscope. Brightness and contrast in images were adjusted uniformly using Photoshop (CS5 or v22). The related primer sequences and genomic coordinates are listed in Supplementary Tables 1 and 5.

PLE elements were PCR-amplified from bacterial artificial chromosomes (Supplementary Table 7) and then cloned into the βlacz plasmid containing a minimal human β-globin promoter-LacZ cassette, as described[49]. Due to their large size, PLE3 (10,351 bp) and PLE5

(9473 bp) were amplified with the proofreading polymerase in the SequalPrep™ Long PCR Kit (Invitrogen). PLE transgenic mice and embryos were produced at the University of Calgary Centre for Mouse Genomics by pronuclear injection of DNA constructs into CD-1 strain single-cell stage embryos[119]. For stable lines, male founder animals (or male F1 progeny produced from transgenic females) were crossed with CD-1 females to produce transgenic embryos which were stained with X-gal by standard techniques[120].

### Generation of Mouse Strains using CRISPR/Cas9

GD$^\Delta$ and SV-Enh$^\Delta$ mouse strains were generated by microinjection or electroporation of CRISPR/Cas9 components into fertilized mouse eggs. Single guide (sg) RNAs located 5' and 3' of the genomic sequence of interest were designed using CHOPCHOP[121] or CRISPOR[122] (http://crispor.tefor.net/), respectively. The GD$^\Delta$ allele was engineered as previously described[123]. Briefly, a mix containing Cas9 mRNA (100 ng/μl) and two single guide RNAs (sgRNAs) (25 ng/μl each) in injection buffer (10 mM Tris, pH 7.5; 0.1 mM EDTA) was microinjected into the cytoplasm of fertilized FVB/NJ (Jackson Laboratory; Strain#:001800) strain oocytes obtained from the oviducts of super-ovulated 7–8 weeks old FVB/NJ females mated to 7–8 weeks old FVB/NJ males. The injected embryos were cultured in M16 medium supplemented with amino acids at 37 °C under 5% CO2 and transferred into the uteri of pseudo-pregnant CD-1 (Charles River Laboratories; Strain Code: 022) surrogate mothers on the same day. The SV-Enh$^\Delta$ allele was engineered using CRISPR-EZ[70] at the Center of Transgenic Models (CTM) of the University of Basel. HiFi Cas9 Nuclease V3 (16 μM) enzyme was incubated with cr:tracrRNA (8 μM each) in a 1:1 molar ratio (IDT) in Hepes-KCl buffer. Minimal Essential Medium (MEM) was added to get a final concentration of 8uM for the electroporation. Following incubation in M16 (Sigma/Merck M7292) with sodium bicarbonate and lactic acid at 37 °C 5%CO2, FVB/NRj (Janvier Labs) strain mouse oocytes obtained from the oviducts of super-ovulated FVB/NRj females (8 weeks) mated to FVB/NRj males (8 weeks or older) were electroporated with the RNP mix. Subsequently, embryos were cultured again in supplemented M16 medium until transferred into the oviduct of pseudo-pregnant Swiss Albino (Janvier Labs; Strain Name: RjOrl:SWISS) females on the same day. CRISPR-derived founder mice (F0) were genotyped using PCR with High Fidelity Platinum Taq Polymerase (Thermo Fisher Scientific) (GD$^\Delta$ line) or conventional Taq Polymerase (SV-Enh$^\Delta$) to identify non-homologous end-joining (NHEJ)-generated deletion breakpoints. Sanger sequencing was used to identify and confirm deletion breakpoints in F0 and F1 mice (see Supplementary Figs. 4 and 5 for genotyping strategy, primers, genotyping PCR and Sanger sequencing).

### Generation of the LHB deletion mouse line

A template allele for genomic deletion of the LHB region[49] encompassing the hs1262/hs1251 enhancers (LHB$^\Delta$) was first produced in G4 mouse embryonic stem cells (mESCs), a hybrid of 129 and C57BL/6 lines[124], at the Centre for Mouse Genomics at the University of Calgary (Supplementary Fig. 7A–D). Briefly, a 11,978 bp genomic fragment (mm10, chr3:66930780-66942757) containing the LHB region was cloned into plasmid pL253[125] from bacterial artificial chromosome (BAC) RP23-213a24 (BACPAC Genomics, Emeryville, California) using gap-repair[126]. For generation of the targeting construct, PCR fragments were amplified from BAC RP23-213a24 and ligated into plasmid pL253 following restriction enzyme digest to replace the genomic 5876 bp LHB region (mm10, chr3:66934220-66940095) with a neomycin (PGK-NEO) selection cassette flanked by LoxP sites using recombineering in E. coli (strain SW102)[125,127,128] (Supplementary Fig. 7A–C). NotI linearized targeting vector was then electroporated into G4 mESCs and clones selected on G418 media were screened for homologous recombination using Southern blotting (SB) with 5' and 3' external probes (Supplementary Fig. 7E), as described[129]. Of 358 clones screened, a single positive clone (#303) was identified to encode deletion of the LHB

region using SB of SacI genomic digests (primary screen), Sph1 genomic DNA digests with a 5' probe, and SacI digests with a 3' probe (Supplementary Fig. 7E). For generation of mouse chimeras, the correctly targeted ES cell clone was aggregated with CD1-strain Morulae and transferred to pseudo pregnant foster females[130]. Six out of seven chimeric male mice were found to transmit the LHB$^\Delta$ allele through the germline to produce heterozygous progeny that were bred to homozygosity. The neomycin selection cassette of the targeted allele was removed in vivo by passing the floxed allele through the germline of Prx1-Cre females[74], yielding the final deletion allele as shown in Supplementary Fig. 7D. Southern blotting and PCR confirmed the in vivo deletion of the LHB region in mice and the latter was used for genotyping with conventional Taq polymerase (Supplementary Fig. 7F). Homozygous LHB$^\Delta$ mice were viable and fertile, without overt phenotypic abnormalities. PCR primers used for recombineering, SB probe amplification and genotyping are listed in Supplementary Table 8.

### ENCODE H3K27ac ChIP-seq and mRNA-seq analysis

To establish a heatmap revealing putative enhancers and their temporal activities within the *Shox2* TAD interval, a previously generated catalog of strong enhancers identified using ChromHMM[56] across mouse development was used[57]. Briefly, calls across 66 different tissue-stage combinations were merged and H3K27ac signals quantified as log2-transformed RPKM. Estimates of statistical significance for these signals were associated to each region for each tissue-stage combination using the corresponding H3K27ac ChIP-seq peak calls. These were downloaded from the ENCODE Data Coordination Center (DCC) (http://www.encodeproject.org/, see Supplementary Data 1, sheet 3 for the complete list of sample identifiers). To this purpose, short reads were aligned to the mm10 assembly of the mouse genome using Bowtie[131], with the following parameters: -a -m 1 -n 2 -l 32 -e 3001. Peak calling was performed using MACS v1.4[132], with the following arguments: --gsize=mm --bw = 300 --nomodel --shiftsize = 100. Experiment-matched input DNA was used as control. Evidence from two biological replicates was combined using IDR (https://www.encodeproject.org/data-standards/terms/). The q-value provided in the replicated peak calls was used to annotate each putative enhancer region defined above. In case of regions overlapping more than one peak, the lowest q-value was used. RNA-seq raw data was downloaded from the ENCODE DCC (http://www.encodeproject.org/, see Supplementary Data 1, sheet 3 for the complete list of sample identifiers). To determine a more permissive set of putative enhancers using less stringent parameters, within the *Shox2* TAD and in major *Shox2* expressing tissues, H3K27ac ChIP-seq peak calling was first run using three different thresholds providing increasingly lower number of peaks (from more to less stringent: $p$-value < 0.00001, $q$-value < 0.05, $p$-value < 0.001) considering midbrain, hindbrain, limb, facial prominence and heart tissues (ENCODE3, E10.5-E15.5 datasets).

### Extended predictions of putative enhancers in the *Shox2* TAD

Peaks resulting from the ENCODE-based analysis described above were used to define and annotate an extended list of putative enhancers in the *Shox2* TAD (Supplementary Data 1, sheets 4 and 5). Briefly, filtering (-q 10) and removal of duplicates was performed using Samtools (v1.14). MACS2 (v2.2.7.1) was used for peak calling. For a given threshold, isogenic replicates were concatenated and further merged ('merge -i') using bedtools (v2.30.0). Genome-wide peaks in the *Shox2* TAD interval (chr3:65996078-67396078) were extracted using the BEDOPS tool (v2.4.39) with the command "bedextract". A master list of putative enhancer regions was first inferred by merging H3K27ac peaks from all stages and tissues, identified at the least stringent threshold ($p$-value < 0.001). The resulting regions were then stitched together if lying within 1 kb from each other, using bedtools merge with "-d 1000". Subsequently, peaks determined at different

thresholds (from more to less stringent: *p*-value < 0.00001, *q*-value < 0.05, *p*-value < 0.001) were used to determine the number of extra putative enhancer regions identified at different stringencies and including also data at E10.5[57]. These regions were further intersected with strong TSS-distal enhancer elements as determined by ChromHMM using the same data[57]. GREAT[133] v4.0.4 was used to re-evaluate which elements were in close proximity (+/− 2.5 kbp) to the TSS of annotated genes.

### Region capture Hi-C (C-HiC)

Embryonic forelimbs (FL), mandibular processes (MD), and hearts (H) from 10 (FL, MD) and 20 (H) wildtype mouse embryos (strain: FVB/NRj) at E11.5 were micro-dissected in cold 1xPBS, pooled according to tissue type, and homogenized using a Dounce tissue grinder. Cells were resuspended in 10% FCS (in PBS) and 1 ml of formaldehyde (37% in H2O, Merck) diluted to a final 2% was added for fixation for 10 min, as previously described[63]. 1.25 M Glycine was used to quench fixation and pellets were snap-frozen in liquid nitrogen and stored at −80 °C. Pellets were resuspended in fresh lysis buffer (10 mM Tris, pH7.5, 10 mM NaCl, 5 mM MgCl2, 0.1 mM EGTA complemented with Protease Inhibitor) for nuclei isolation. Following 10 min incubation on ice, samples were washed with 1xPBS and frozen in liquid nitrogen. 3C-libraries were prepared from thawed nuclei subjected to DpnII digestion (NEB, R0543M), re-ligated with T4 ligase (Thermo Fisher Scientific) and de-crosslinking as described previously[63]. For 3C-library quality control, 500 ng of library sample along with digested and undigested control samples was assessed using agarose gel electrophoresis (1% gel). Shearing on re-ligated products was performed using a Covaris ultra-sonicator (duty cycle: 10%, intensity 5, cycles per burst: 200, time: 2 cycles of 60 s each). Following adaptor ligation and amplification of sheared DNA fragments, libraries were hybridized to custom-designed SureSelect beads (SureSelectXT Custom 0.5–2.9 Mb library) and indexed following Agilent's instructions. Multiplexed libraries were sequenced using 50 bp paired-end sequencing (HiSeq 4000 sequencer). C-HiC probes of the SureSelect library were designed to span the *Shox2* genomic interval and adjacent TADs (mm10: chr3:65196079-68696078).

### C-HiC data processing and analysis

C-HiC processing was performed using a previously published pipeline[28]. Briefly, sequenced reads were mapped to the reference genome GRCm38/mm10 following the HiCUP pipeline[134] (v0.8.1) set up with Bowtie2[135] (v2.4.5). Filtering and de-duplication was conducted using HiCUP (no size selection, Nofill: 1, format: Sanger) and unique MAPQ ≥ 30 valid read pairs were obtained for FL, MD and HT datasets ($N = 637{,}163$, $N = 577{,}862$ and $N = 592498$, respectively). Binned contact maps from valid read pairs were generated using Juicer command line tools[136] (v1.9.9) and raw.cool files were generated with the hicConvertFormat tool (HiCExplorer v3.7.2) from native .hic out-puts generated by Juicer. For normalization and diagonal filtering the Cooler matrix balancing tool[137] (v0.8.11) was applied with the options '--mad-max 5 --min-nnz 10 --min-count 0 --ignore-diags 2 --tol 1e-05 --max-iters 200 --cis-only'. Only the targeted genomic interval enriched in the capture step (mm10: chr3:65196079-68696078) was selected for binning and balancing. Consequently, only read pairs mapping to this interval were retained, shifted by the offset of 65,196,078 bp using custom chrome.sizes files. Balanced maps were then exported at 5 kb resolution with corrected coordinates (transformed back to original values). Subtraction maps were directly generated from Cooler balanced Hi-C maps using the hicCompareMatrices tool (HiCExplorer v3.7.2) with option '--operation diff'. HiCExplorer[138] (v3.7.2) was used to determine normalized inter-domain insulation scores and domain boundaries on Hi-C and subtraction maps using default parameters 'hicFindTADs -t 0.05 -d 0.01 -c fdr' computing *p*-values for a minimal window length of 50000. Hi-C maps and related graphs were visualized from.cool files and bedgraph matrices, respectively, using pyGenomeTracks[139] (*v.3.6*). GOTHiC[140] (v.1.32.0) was used to identify reliable and significance-based Hi-C interactions from HiCUP validated read pairs (MAPQ10) with 'res=1000, restrictionFile, cistrans = 'all', parallel=FALSE, cores=NULL' (R pipeline-template script, v.4.2.2) and a threshold of '-log(q-value) > 1'.

### Virtual 4C (v4C)

To determine target interactions of a defined element locally v4C profiles were generated as described[63] from filtered unique read pairs (hicup.bam files) which also served as input for computation of C-HiC maps (see above). Conditions for mapped read-pairs included MAPQ ≥ 30 and relative position of the two reads inside and outside the viewpoint, respectively. After quantitation of reads outside of the viewpoint (per restriction fragment), read counts were distributed into 3 kb bins (with proportional distribution of read counts in case of overlap with more than one bin). Following smoothing of each binned profile via averaging[63], peak profiles were generated using custom Java code based on htsjdk v2.12.0 (https://samtools.github.io/htsjdk/). A 10 kb viewpoint containing the extended *Shox2* promoter region (chr3:66975788-66985788) was used for comparison with Hi-C maps. The viewpoint and neighboring +/-5 kb regions were excluded from computation of the scaling factor. BigwigCompare tool (deepTools v3.5.1) was used to generate relative subtraction Capture-C-like profiles.

### 4C-seq from proximal forelimbs

10–12 proximal forelimbs from CD-1 embryos at E12.5 were dissected per biological replicate sample ($n = 2$ in total) in PBS, followed by 4C-seq tissue processing as described[141,142]. For tissue preparation, cells were dissociated by incubating the pooled tissue in 250 µl PBS supplemented with 10% fetal calf serum (FCS) and 1 mg/ml collagenase (Sigma) for 45 min at 37 °C with shaking at 750 rpm. The solution was passed through a cell strainer (Falcon) to obtain single cells which were fixed in 9.8 ml of 2% formaldehyde in PBS/10% FCS for 10 min at room temperature and lysed. Libraries were prepared by overnight digestion with NlaIII (New England Biolabs (NEB)) and ligation for 4.5 hours with 100 units T4 DNA ligase (Promega, #M1794) under diluted conditions (7 ml), followed by de-crosslinking overnight at 65 °C after addition of 15 ul of 20 mg/ml proteinase K. After phenol/chloroform extraction and ethanol precipitation the samples were digested overnight with the secondary enzyme DpnII (NEB) followed by phenol/chloroform extraction, ethanol precipitation purification and ligation for 4.5 h in a 14 ml volume. The final ligation products were extracted and precipitated as above followed by purification using Qiagen nucleotide removal columns. For each viewpoint, libraries were prepared with 100 ng of template in each of 16 separate PCR reactions using the Roche, Expand Long Template kit with primers incorporating Illumina adapters. Viewpoint and primer details are presented in Supplementary Table 6. PCR reactions for each viewpoint were pooled and purified with the Qiagen PCR purification kit and sequenced with the Illumina HiSeq to generate single 100 bp reads. Demultiplexed reads were mapped and analyzed with the 4C-seq module of the HTSstation pipeline as described[143]. Results are shown in UCSC browser format as normalized reads per fragment after smoothing with an 11-fragment window and mapped to mm10 (Fig. 6B, Supplementary Fig. 6E). Raw and processed (bedgraph) sequence files are available under GEO accession number GSE161194.

### Whole-mount in situ hybridization (ISH)

For assessment of spatial gene expression changes in mouse embryos, whole mount in situ hybridization (ISH) using a *Shox2* digoxigenin-labeled antisense riboprobe[21] was performed as previously described[144]. Briefly, embryos were fixed in 4% paraformaldehyde (PFA) in PBS at 4 °C overnight, dehydrated through a 25%/50%/75%

methanol/PBT series and stored in 100% methanol at −20 °C until further processing. Following rehydration in a reverse methanol/PBT series, embryos were bleached in 6% hydrogen peroxide (in PBT) for 15 min and then digested with 10 μg/ml proteinase K (20 min for E10.5, 25 minutes for E11.5). After PK permeabilization, samples were treated with freshly prepared 2 mg/ml glycine in PBT for 5 minutes and post-fixed in 0.2% glutaraldehyde/4%PFA in PBT for 20 min. Following incubation in pre-hybridization buffer (50% deionized formamide; 5x SSC pH 4.5; 2% Roche Blocking Reagent; 0.1% Tween-20; 0.5% CHAPS; 50 mg/mL yeast RNA; 5 mM EDTA; 50 mg/ml heparin) at 65 °C ($\geq$ 3 h), embryos were incubated overnight in 1 ml of hybridization solution containing 1 μg/ml DIG-labeled *Shox2* riboprobe at 70 °C. The next day, embryos were extensively washed and non-hybridized riboprobe was digested by 20 μg/ml RNase for 45 min at 37 °C. After additional washes and pre-blocking, the embryos were incubated overnight with anti-digoxigenin antibody (1:5000, Roche cat. no. 11093274910) at 4 °C. Following extensive washing to remove excess antibodies and equilibration in NTMT, the mRNA signal was developed by incubation in BM purple (Roche cat. no. 11442074001) and stopped before saturation by several washes in PBT. For comparative analysis between genotypes, incubation in BM purple was conducted for the same period per embryonic stage. Whole-mount ISH analyses in embryos are qualitative and well suited to detect spatial changes. At least $n = 3$ independent embryos were analyzed for each genotype. Embryonic tissues were imaged using a Leica MZ16 microscope coupled to a Leica DFC420 digital camera. Brightness and contrast were adjusted uniformly using Photoshop (CS5).

## Quantitative real-time PCR (qPCR)

Mouse embryonic limb buds, hearts and craniofacial compartments at E10.5-E11.5 were micro-dissected in ice-cold PBS, transferred to RNA-later (Sigma-Aldrich) and stored at −20 °C until further use. Dissected limb buds collected for experiments focused on the LHB$^\Delta$ allele were additionally homogenized with a Qiagen tissueruptor II. For qPCR experiments focused on the GD$^\Delta$ allele, isolation of RNA from micro-dissected embryonic tissues was performed using the Ambion RNA-queous Total RNA Isolation Kit (Life Technologies) according to the manufacturer's protocol. For qPCR experiments focused on SV-Enh$^\Delta$ and LHB$^\Delta$ alleles, RNeasy Micro and Mini Kits (Qiagen) were used, respectively. RNA was reverse transcribed using SuperScript III (Life Technologies) with poly-dT (GD$^\Delta$ and SV-Enh$^\Delta$) or random hexamer (LHB$^\Delta$) priming. For GD$^\Delta$ samples, qPCR was conducted on a Light-Cycler 480 (Roche) using KAPA SYBR FAST qPCR Master Mix (Kapa Biosystems). For SV-Enh$^\Delta$ samples, a ViiA 7 Real-Time PCR System using PowerTrack SYBR Green Master Mix (Applied Biosystems) was used. qPCR for LHB$^\Delta$ samples was performed on a Quantstudio 4 (Applied Biosystems) using the PowerUP SYBR Green Master Mix (Applied Biosystems). All primers used for qPCR were described previously[21] (Supplementary Table 6). Relative quantification of transcripts was calculated using the $2^{-\Delta\Delta C_T}$ method (GD$^\Delta$ and SV-Enh$^\Delta$)[21] or using the efficiency correction method and comparison to a 6-point standard curve for each primer pair[145] (LBH$^\Delta$) and normalized to the *Actb* housekeeping gene. The mean of wild-type control samples was set to 1, as used previously[21]. Tissues from at least $n = 5$ embryos (biological replicates) were analyzed per genotype.

## Immunofluorescence (IF)

IF was performed as previously described[21]. Briefly, mouse embryos at E11.5 were isolated in cold PBS and fixed in 4% PFA for 2–3 h. After incubation in a sucrose gradient and embedding in a 1:1 mixture of 30% sucrose and OCT compound, sagittal 10μm frozen tissue sections were obtained using a cryostat. Selected cryo-sections were blocked using BSA and incubated overnight with the following primary antibodies: anti-SHOX2 (1:300, Santa Cruz JK-6E, sc-81955), anti-HCN4 (1:500, Thermo Tn Fisher, MA3-903) and anti-NKX2.5 (1:500, Thermo Fisher,

PA5-81452). Sections were incubated for 1 h in a mix of donkey anti-mouse Alexa Fluor 647 (1:1000, Thermo Fisher, # A31571), goat anti-rat 568 Alexa Fluor (1:1000, Thermo Fisher, #A11077) and goat anti-rabbit 488 (1:1000, Thermo Fisher, #A11008) secondary antibodies for detection. For Supplementary Fig. 4D, sections were incubated with anti-SMA-Cy3 for 1 h (1:250, Sigma, #C6198) following treatment with anti-SHOX2 and anti-mouse Alexa Fluor 647, as described above. Hoechst 33258 (Sigma-Aldrich) was utilized to counterstain nuclei. A Zeiss AxioImager fluorescence microscope in combination with a Hamamatsu Orca-03 camera was used to acquire fluorescent images. Brightness and contrast were adjusted uniformly using Photoshop (CS5). Three biological replicates (embryos) were analyzed for GD$^{\Delta/\Delta}$ and two for wildtype control genotypes.

## Skeletal preparations

For limb skeletal preparations, newborn mice were euthanized at P0 and subsequently eviscerated, skinned and fixed in 1 % acetic acid in EtOH for 24 h. Cartilage was stained overnight with 1 mg/mL Alcian blue 8GX (Sigma) in 20% acetic acid in EtOH. After washing in EtOH for 12 h and treatment with 1.5% KOH for three hours, bones were stained in 0.15 mg/mL Alizarin Red S (Sigma) in 0.5% (w/v) KOH for four hours and cleared in 20% glycerol, 0.5 % KOH. Fore- and hindlimbs of at least $n = 4$ biological replicates were analyzed for control genotypes and at least $n = 7$ for the GD$^\Delta$/Shox2$^{\Delta c}$ genotype. Stained P0 skeletons were blinded and randomized prior to measuring. Disarticulated bones of the right limbs were measured manually under a Leica MZ 125 dissecting microscope using an electronic digital caliper (Fine Science Tools, Catalog #30087-00). The length of the humerus and femur are reported as the average of three blinded measurements to improve precision and reduce error. The lengths of the humerus and femur were normalized to the length of the third metatarsal, where *Shox2* is not expressed.

## X-ray micro-computed tomography (μCT) of adult mouse skeletons

Mice were euthanized at 6 weeks of age and whole-body μCT scans were generated using a Skyscan 1173 v1.6 μCT scanner (Bruker, Kontich, Belgium) at 80–85 kV and 56–62 μA with 45 μm resolution[146]. NRecon v1.7.4.2 (Bruker, Kontich, Belgium) was used to perform stack reconstructions and 3D landmarks were placed in MeshLab[147] (v2020.07) by one observer (CSS) blind to the genotype identity of individual animals. Limb skeletons from at least $n = 4$ biological replicates were measured for control genotypes and at least $n = 8$ for GD$^\Delta$/ Shox2$^{\Delta c}$ genotypes. To quantify the length of the stylopod bones, distances were calculated between two landmarks placed at the proximal and distal ends of the humerus and femur (the proximal epiphysis [PE] and olecranon fossa lateral [OFL] for the humerus, and the greater trochanter [GT] and lateral inferior condyle [LIC] for the femur). To account for body size variability between individuals, these measurements were normalized to the inter-landmark distance between the proximal and distal ends of the third metatarsal. To assess intra-observer repeatability, CSS placed the landmarks on scans of 12 mice (six GD$^\Delta$/Shox2$^{\Delta c}$, two GD$^{\Delta/+}$, two Shox2$^{\Delta c/+}$, and two WT) five times each, with each session separated by at least 24 hours[148]. An absolute coefficient of variation (CV) for each landmark was calculated and the average CV was 0.28% with a range of 0.14–0.42%.

## ATAC-seq

ATAC-seq was performed as described[149] with minor modifications. Per biological replicate ($n = 2$ in total), pairs of wildtype mouse embryonic hearts at E11.5 were micro-dissected in cold PBS and cell nuclei were dissociated in Lysis buffer using a Dounce tissue grinder. Approx. 50'000 nuclei were then pelleted at 500 RCF for 10 min at 4 °C and resuspended in 50 μL transposition reaction mix containing 25 μL Nextera 2x TD buffer and 2.5 μL TDE1 (Nextera Tn5 Transposase;

Illumina) (cat. no. FC-121-1030) followed by incubation for 30 min at 37 °C with shaking. The reaction was purified using the Qiagen MinElute PCR purification kit and amplified using defined PCR primers[150]. ATAC-seq libraries were purified using the Qiagen MinElute PCR purification kit (ID: 28004), quantified by the Qubit Fluorometer with the dsDNA HS Assay Kit (Life Technologies) and quality assessed using the Agilent Bioanalyzer high sensitivity DNA analysis assay. Libraries were pooled and sequenced using single end 50 bp reads on a HiSeq 4000 (Illumina).

### Mouse ATAC-seq and ChIP-seq data processing

Analysis of heart ATAC-seq (E11.5) and reprocessing of previously published ATAC-seq and ChIP-seq datasets used in this study (see Supplementary Data 3) was performed using Adaptor trimming (trim_galore_v0.6.6) by Cutadapt (https://cutadapt.readthedocs.io/), with default parameters '-j 1 -e 0.1 -q 20 -O 1' for single-end, and '--paired -j 1 -e 0.1 -q 20 -O 1' for paired- end data (purging trimmed reads shorter than 20 bp). For read mapping, Bowtie2[135] (version 2.4.2) was used with parameters '-q --no-unal -p 8 -X2000' (ATAC-seq) and '-q --no-unal -p 2' (ChIP-seq) for both single/paired-end samples. Reads were aligned to the GRCm38/mm10 reference genome using pre-built Bowtie2 indexes from the Illumina's iGenomes collection (http://bowtie-bio.sourceforge.net/bowtie2/). Duplicates and low-quality reads (MAPQ = 255) for both single/paired-end samples were removed using SAMtools (v1.12), with pipeline parameters 'markdup -r' and '-bh -q10', respectively[151]. ATAC-seq peak calling was performed using MACS2[132,152] (v2.1.0) with p-value < 0.01 and parameters '-t -n -f BAM -g mm --nolambda --nomodel --shif 50 --extsize 100' for single-end, and '-t -n -f BAMPE -g mm --nolambda --nomodel --shif 50 --extsize 100' for paired-end reads. For ChIP-seq peak calling, '-t -c -n -f BAM -g mm' parameters were used instead. PyGenomeTracks[139] was used for visualization of profiles and alignment with other datasets.

### Cardiac TF motif detection

An enriched collection of position weight matrices (PWMs)[153] was limited to motifs of TFs expressed in the developing heart at E11.5. After mapping of gene symbols to the equivalent identifiers in the Ensembl103 release using the BiomaRt v2.5.0 package (R v4.1.2)[154], only those PWMs matching TFs expressed in E11.5 hearts were selected for analysis[155] (ENCSR691OPQ). A mean FPKM ≥ 2 calculated across all RNA-seq replicates was used as threshold for significant expression. This filtering resulted in a set of 576 mouse TFs. 1'376 corresponding PWMs were available for 282 of these TFs[69] which were used for motif detection by FIMO (Find Individual Motif Occurrences)[69,156], except for 14 that were omitted since in each case, since the match identified genome-wide was included in a larger motif within the collection (Supplementary Data 4). FIMO v5.3.0 with a standard p-value cutoff of $10^{-4}$ and GC-content matched backgrounds was used for screening genomic sequence for potential TF-binding sites. Motif conservation was computed using BWTOOL v1.0[157] based on the average of individual nucleotide PhyloP (Placental) conservation scores provided by UCSC PHAST package (http://hgdownload.cse.ucsc.edu/goldenpath/mm10/phyloP60way/).

### ChIP-seq and RNA-seq from human fetal hearts

Fetal human RV, LA and RA tissue samples at post conception week (PCW17) obtained from the Human Developmental Biology Resource's Newcastle site (HDBR, hdbr.org) were transported on dry ice, stored at −80 °C and processed for ChIP-seq and RNA-seq analogous to the procedure for the fetal LV sample of the same origin[115]. ChIP-seq libraries were prepared using the Illumina TruSeq library preparation kit, and pooled and sequenced (50 bp single end) using a HiSeq2000 (Illumina). Processing was performed using a previously published pipeline[21], with minor modifications. Briefly, ChIP-seq reads were obtained following quality filtering and adaptor trimming using

cutadapt_v1.1 with parameter '-m 25 -q 20'. Bowtie[131] (v2.0.2.0) with parameter '-m 1 -v 2 -p 16' and MACS[132] (v1.4.2) with parameter '-mfold = 10,30 -nomodel -p 0.0001' were used for read mapping (hg19) and peak calling, respectively. Duplicates were removed with SAMtools[151]. RNA-seq libraries were prepared using the TruSeq Stranded Total RNA with Ribo-Zero Human/Mouse/Rat kit (Illumina) according to manufacturer instructions. An additional purification step was used to remove remaining high molecular weight products, as published[115]. RNAseq libraries were pooled and sequenced via single end 50 bp reads on a HiSeq 4000 (Illumina) and processed as previously published, with minor modifications[21]. Briefly, RNA-seq reads were pre-processed using quality filtering and adaptor trimming with cutadapt_v1.1 ('-m 25 -q 25'). Tophat v2.0.6 was used to align RNA-seq reads to the mouse reference genome (hg19) and the reads mapping to UCSC known genes were determined by HTSeq[158] (v0.7.0). Normalized bigWig files were generated using bedtools (bedGraphToBigWig) and IGV browser was used for visualization of profiles.

### Statistics and reproducibility

Statistical analyses are described in detail in the Methods section above. For fetal human heart samples, cardiac compartments (LV, RV, LA, RA) from only one human embryo (XY) at post conception week 17 were analyzed for ChIP-seq and RNA-seq. Results from transient transgenic enhancer analysis reported in this study results were confirmed in at least two (enSERT) or three (Hsp68 random integration) independent embryos (biological replicates) based on criteria consistent with results established at LBNL for the VISTA Enhancer Browser (http://enhancer.lbl.gov). For experiments focused on genomic deletion alleles, sample sizes were selected based on our previous studies[21,22] and per experiment the minimal number of biological replicates determined is listed in the respective Methods sections. Individuals who qualitatively assessed the results of in vivo transgenic reporter assays or measured skeletal elements were blinded to genotyping information. For all other experiments, the investigators were not blinded to allocation during experiments and outcome assessment. No statistical method was used to pre-determine sample size. No data that passed quality control criteria for experiments were excluded from the analyses. The experiments were not randomized. Unless otherwise stated, default parameter settings were used for any software tool employed in the analyses. Whenever a p-value is reported in the text or figures, the statistical test is also indicated. μCT measurement plots were generated and statistically analyzed with GraphPad Prism version 10.2.3. All other statistics were estimated, and plots were generated using the statistical computing environment R version 4.3.2.

### Reporting summary

Further information on research design is available in the Nature Portfolio Reporting Summary linked to this article.

## Data availability

The raw and processed next-generation sequencing (NGS) datasets generated in this study have been deposited in the NCBI GEO database under accession codes GSE161194 (4C-seq) and GSE232887 (superseries including C-HiC (GSM7385429-30), ATAC-seq (GSM7385432-33), ChIP-seq (GSM7385434-41) and RNA-seq data (GSM7385442-45)). Accession codes of previously published ATAC-seq (GSE124338 [https://www.ncbi.nlm.nih.gov/geo/query/acc.cgi?acc=GSE124338][66], GSE148515[46], GSE126293[144]) and ChIP-seq (GSE96107 [https://www.ncbi.nlm.nih.gov/geo/query/acc.cgi?acc=GSE96107][64]; GSE137285[159]; GSE124008[67], GSE52123[68], GSE68974[160], GSE123388[63], GSE129427[59]; ENCODE[58]: ENCFF310VOQ, ENCFF464DYI) datasets reprocessed in this study are listed in Supplementary Data 3 with the respective NarrowPeak files are available in Supplementary Data 5. Wherever applicable, reference genomes Mouse GRCm38/mm10 and Human GRCh37/hg19 were used for alignment and comparisons. Images of transgenic

embryos with LacZ-reporter activity are available at the Vista Enhancer Browser (http://enhancer.lbl.gov). Source data are provided with this paper. Correspondence and requests for materials should be addressed to J.C. (jacobb@ucalgary.ca) or M.O. (marco.osterwalder@unibe.ch). Source data are provided with this paper.

## Code availability

This study made use of current community-accepted and benchmarked bioinformatic analysis methods which are cited in the main text or Methods section. No previously unreported custom computer code, mathematical or software algorithms were used for data analysis.

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

## Acknowledgements

We thank L. Lopez-Delisle for sharing expertise on the use of pyGenomeTracks and Capture Hi-C analysis, G. Kelman for preliminary ATAC-seq analysis and D. Duboule for hosting and supporting 4C-Seq experiments as well as training in his laboratory. We are grateful to P. Pelczar and the members of the University of Basel Center for Transgenic Models (CTM) for generation of the mouse SV-Enh^Δ deletion allele and thank C. Detotto and her team at the Central Animal Facilities (CAF) of the University of Bern for excellent mouse care. We are grateful to M. Docquier and her team from the iGE3 facility for preparation and sequencing of C-HiC libraries. We thank C. Fielding at the Clara Christie Centre for Mouse Genomics for pronuclear injections conducted at the University of Calgary, J. Theodor and J. Anderson for use of the SkyScan 1173 uCT scanner and C. Rolian and C. Unger for help with morphometric analysis. We thank V. Rapp for cloning the +325 transgenic reporter construct. We are grateful to the members of the L.A.P. and A.V. group for technical advice and the members of the M.O. and J.C. labs for useful comments on the manuscript. This work was supported by Swiss National Science Foundation (SNSF) grant PCEFP3_186993 (to M.O.), Discovery Grants (RGPIN-2013-355731 and RGPIN-2019-04812) from the Natural Sciences and Engineering Research Council of Canada (to J.C.) and National Institutes of Health grants R01HG003988, U54HG006997, R24HL123879 and UM1HL098166 (to A.V. and L.A.P.). M.O. was also supported by grants of the Swiss Heart Foundation (FF20110) and Novartis Foundation for Medical-Biological Research (#21C183). J.L-R. is supported by the MICINN grants PID2020-113497GB-I00 and CEX2020-001088-M (Unidad de Excelencia María de Maeztu institutional grant). G.A. is supported by Swiss National Science Foundation Grants PP00P3_176802 and PP00P3_210996. F.D. is supported by a SNSF postdoc.mobility fellowship (P400PB_194334). Research at the E.O. Lawrence Berkeley National Laboratory was performed under Department of Energy Contract DE-AC02-05CH11231, University of California.

## Author contributions

M.O. and J.C. conceived the study. S.A.-O., M.Z. and B.J.M performed critical experimental (S.A.O., B.J.M.) and computational (M.Z.) analyses for the study. S.A.-O., B.J.M., M.K., J.C. and M.O. designed and performed transgenic reporter and gene expression analyses. R.R. conducted experimental C-HiC. V.T. and J.L.-R. executed the in-situ hybridization analysis. M.Z. performed C-HiC and ATAC-seq/ChIP-seq data processing and analysis from all mouse datasets. I.B. set up the enhancer profiling framework based on ENCODE data and ChromHMM. C.H.S. and B.J.M. conducted ChIP-seq and RNA-seq from human heart tissues. Y. F.-Y. performed ChIP-seq and RNA-seq processing and analysis of human heart datasets. S.A.-O., E.R.-C., A.I., G.A. and J.C. performed 4C-seq experiments and analysis. V.R and J.G. conducted SV

enhancer-deletion experiments. F.D., A.I., R.H., J.A.A. performed additional experimental work related to transgenic reporter validation. T.A.F and C.S.S. did skeletal phenotyping. C.S.N, I.P.-F. and S.T. performed pro-nuclear injections. G.A., D.E.D., A.V. and L.A.P. provided project funding and support. J.C. and M.O. provided project funding and wrote the manuscript with input from the other authors.

## Competing interests

The authors declare no competing interests.

## Additional information

[1]Department of Biological Sciences, University of Calgary, 2500 University Drive N.W., Calgary, AB T2N 1N4, Canada. [2]Department for BioMedical Research (DBMR), University of Bern, 3008 Bern, Switzerland. [3]Environmental Genomics and Systems Biology Division, Lawrence Berkeley National Laboratory, Berkeley, CA 94720, USA. [4]Comparative Biochemistry Program, University of California, Berkeley, CA 94720, USA. [5]Department of Genetic Medicine and Development and iGE3, Faculty of Medicine, University of Geneva, Geneva, Switzerland. [6]Centro Andaluz de Biología del Desarrollo (CABD), CSIC-Universidad Pablo de Olavide-Junta de Andalucía, 41013 Seville, Spain. [7]Department of Cardiology, Bern University Hospital, 3010 Bern, Switzerland. [8]Department of Genetics and Evolution, University of Geneva, Geneva, Switzerland. [9]School of Health Sciences, Universidad Loyola Andalucía, Seville, Spain. [10]Center for Cancer Research, Medical University of Vienna, Vienna, Austria. [11]US Department of Energy Joint Genome Institute, Lawrence Berkeley National Laboratory, Berkeley, CA 94720, USA. [12]School of Natural Sciences, University of California, Merced, Merced, CA 95343, USA. [13]Present address: Department of Biological Sciences, Fort Hays State University, Hays, KS 67601, USA. [14]Present address: Department of Molecular Biology, University of Geneva, Geneva, Switzerland. [15]These authors contributed equally: Samuel Abassah-Oppong, Matteo Zoia, Brandon J. Mannion. ✉e-mail: jacobb@ucalgary.ca; marco.osterwalder@unibe.ch

