## [Peer Review File · Nature Communications]

REVIEWER COMMENTS

Reviewer #1 (Remarks to the Author):

In this paper, Abassah-Oppong et al., investigate the role of the gene desert flanking the *Shox2* locus in its transcriptional regulation. *Shox2* encode a transcription factor involved in limb, craniofacial, hindbrain and cardiac morphogenesis. The authors first did a screen to identify putative active enhancer elements in the gene desert. The in vivo activity patterns of these putative enhancer elements were then assessed by performing LacZ transgenic reporter assays in mouse embryos. In total, the authors tested 25 individual enhancer elements (DE1 to DE16, +319kb, +325kb, +389kb, +405kb, +417kb, +515kb, +520kb; PEL3 and PLE5; PLE4/DE4 being the same enhancer element). While some of the tested enhancers displayed no staining (i.e. +319kb, +389kb, +405kb, +417kb, +515kb, +520kb enhancers) or no reproducible staining, the authors also identified enhancers with specific subregional activities, predominantly in limb, craniofacial, neuronal, and cardiac cell populations, i.e. in regions known to be dependent on *Shox2* expression.

To evaluate the function of the gene desert flanking *Shox2*, the authors deleted it using a CRISPR/cas9 system in mouse. They found that the gene desert interval is important for *Shox2* transcriptional control in developing limbs, craniofacial compartments, and in the cardiac sinus venosus. Phenotypically, the authors show that deletion of the gene desert leads to pacemaker-related embryonic lethality due to *Shox2* depletion in the cardiac sinus venosus. They also report a requirement of the gene desert for stylopod morphogenesis. The authors then conclude that the *Shox2* gene desert can be considered as a fundamental genomic unit that controls pleiotropic *Shox2* expression.

The paper is well written, the presented experiments are of high quality and convincing. The amount of work involved in this study is substantial and the approach thorough and highly informative.

On the other hand, some data are not completely novel. Among the four limb enhancers identified in the gene desert, one (i.e. hs1262/LHB-A) was already described. Moreover, the heart enhancer called SV enhancer identified in the present study is part of a broader region (241kb) called VS-250. Deletion of this region was shown to be sufficient to deplete *Shox2* in the cardiac sinus venosus (SV), resulting in a hypoplastic SAN and abnormally developed venous valve primordia responsible for embryonic lethality. It should however be noted that an effort was made by the authors to refine the analysis, since the SV-enhancer identified here represent 4kb within the VS-250 region of 241kb. By inactivating the SV-enhancer, the authors also validated the role of this enhancer in regulating *Shox2* expression in SV; inactivation of the SV-enhancer was however not sufficient to recapitulate the phenotype obtained by the VS-250 deletion, indicating additional roles of other enhancers.

Overall, I believe this elegant study will be of great interest to the field, provided that the following revisions are implemented.

Main points of revision:

1. While the C-HiC profiles across the *Shox2* TAD and flanking regions are largely similar in mouse embryonic forelimbs, mandibles, and hearts at E11.5, the authors report the presence of a cardiac specific high-density contact domain (HCD) spanning ~170kb in the most distal D-dom compartment. In the model shown in Figure 7, the authors proposed that this HCD may influence heart-specific *Shox2* expression by sequestering non-heart enhancers from interacting with the *Shox2* promoter. However, the role of this cardiac specific HCD is unclear. Indeed, from the figure 1A, it seems that the cardiac enhancer +325kb (which is close to DE3) is interacting with *Shox2* not only in the heart but also in the mandible and in the forelimb, then questioning the role of HCD in allowing the specific interactions between the cardiac enhancer +325kb and *Shox2* in the heart. To clarify this point, the authors should perform a 4C experiment in heart, mandible, and forelimb tissues with a viewpoint on the enhancer +325kb. It is moreover not clear if the enhancer +325kb is specific for heart tissue, or if it is also active in the craniofacial region or in the limbs? Indeed, in the Figure 4E, the LacZ staining is focused on the heart. The authors should clarify this point as well.

2. In the recent study showing that the deletion of VS-250 within the gene desert is sufficient to deplete *Shox2* in the SV (van Eif et al., 2020), the authors also reported the presence of a regulatory element called R4, outside SV-250, that is highly conserved in mammals and able to drive reporter activity in the *Hcn4*+/*Shox2*+ SAN of E11.5 mouse embryos. Was this enhancer identified by the first screening analysis performed by the authors? At any rate, it would be important for completeness to cite it as a putative identified SV enhancer and to display it, in figure 1A, with the other identified *Shox2* enhancers in the gene desert.

More generally, it can be helpful to systematically (on all figures) place the already identified enhancers, in addition to the ones identified here, on the gene desert genomic coordinates.

3. The authors mention that: “Neither knockout of the *hs1262* proximal limb enhancer nor the identification of new limb enhancers (DE6, DE10) located in the gene desert (Fig. 1) was sufficient to explain the ~50% *Shox2* reduction observed in proximal fore- (FL) and hindlimbs (HL) of GDD/D embryos”; however, the reduction of *Shox2* expression in GDD/*Shox2*Dc mutants, as compared to the reduction of *Shox2* expression resulting from *hs1262* alone is not shown. The role and added value of PLE3-5 enhancers on the transcriptional regulation of *Shox2* as compared to *hs1262* has not been evaluated. Also, can the authors clarify whether, in their analysis to identify putative *Shox2* active limb enhancers, they found the previously described limb enhancer *hs638*.

4. To estimate the number of developmental enhancers in the gene desert, the authors established a map of stringent enhancer activities by using ChromHMM-filtered H3K27ac signatures from bulk tissues across 66 embryonic and perinatal tissue-stage combinations, from ENCODE. The time points included in the analysis were E11.5 -15.5. However, the validity of such an approach, even for initial screening, is questionable, given the risk of missing essential enhancers. Indeed, neither the SV enhancer +325kb, nor the limb enhancers PLE3 and PLE5 were identified using this method. The authors moreover mention that “the previously validated hs741 and hs1262 limb enhancers were not among the predictions as elevated H3K27ac marks are present at these enhancers already at earlier stages.” However, as shown in the vista browser and in figure 1A, hs741 and hs1262 are still active at E11.5 (a stage included in the analysis). Shouldn't they then have been identified by the ChromHMM-filtered H3K27ac signature analysis? It would have been more appropriate to dissect mouse tissues at appropriate stages, where *Shox2* is known to be active, and perform H3K27ac ChIPseq +/- , ATACseq +/- , and chromatin conformation capture (C-HiC, 4C-seq). This is what the authors finally did, but only for limbs. It is regrettable that a comparable methodical approach was not taken for the craniofacial and cardiovascular domains.

5. The presented heatmap in Figure 1B is misleading as it suggests that the gene desert enhancers DE1 to DE16 completely cover the gene desert, with each enhancer element being of the same size. However, this is not the case. For clarity, it would be helpful to display boxes on the genomic coordinate of the gene desert presented in Figure 1A, indicating the location and size of each enhancer element.

6. In their analysis using the LacZ transgenic reporter on DE1 to DE16, the authors noted reproducible activity in 12 out of 16 cases tested. Are these 12 cases those depicted in figure 1C? If so, does this imply that DE4/PLE4 (not displayed in Figure 1C at E11.5, but visible in Figure 6C at E12.5) does not exhibit a repeatable pattern? The authors should clarify this issue and specifically mention which enhancer(s) among the DE1-DE16 do not result in a repeatable pattern.

7. The authors mention “Other prominent 4C-seq peaks in the gene desert co-localized with either previously validated enhancer elements with non-limb activities at E11.5 (DE1, 4, 6, 9, 15) or non-validated elements with proximal limb-specific H3K27ac enrichment (+237kb and 346 +568kb)”. The LacZ staining for DE4 at E11.5 is however not shown; it is not clear if DE4 is among the DEs that do not show a reproducible pattern at E11.5 (see also above).

8. The authors state that: “Open chromatin and H3K27ac profiles further indicated that the *Shox2*-interacting DE4 (+407) element was unique in its H3K27ac pattern (initiated at E11.5), while other proximal limb (candidate) enhancers showed activity marks already at E10.5.”

They should refer to the figure where they show this. Indeed, the authors mention that they “reprocessed previously published ChIP-seq datasets from dissected proximal and distal limbs at

E12”; but they never mentioned that they analyzed open chromatin and H3K27ac profiles at earlier stages.

9. The authors state that: “Open chromatin and H3K27ac profiles further indicated that the Shox2-interacting DE4 (+407) element was unique in its H3K27ac pattern (initiated at E11.5), while other proximal limb (candidate) enhancers showed activity marks already at E10.5.” Then, it is mentioned that “...PLE4 (+407) drove reporter activity already at E10.5 in a more widespread pattern leaking into distal forelimbs at later stages, in line with H3K27ac enrichment in distal forelimbs (Figs. 6A, S6A).”

The authors may explain this discrepancy between the time when H3K27ac starts to be observed (E11.5) and the time at which the reporter is observed (E10.5) in the LacZ transgenic reporter assay.

Minor points:

a. The authors state that “... .Despite such critical roles, the functional requirement of only few gene deserts have been studied in detail, including the investigation of chromatin topology and functional enhancer landscapes in the TADs of other developmental key regulators such as Sox9, Shh or Fgf810–12.”. For the sake of completeness, an additional recent study could be cited here that identified functional long-range enhancers in a gene desert involved in the regulation of the key craniofacial regulator Hoxa2 (Kessler et al., Nat Comm 2023).

b. In the following sentence: “While a subset of these candidate SAN enhancer elements overlapped DEs validated for non-cardiac activities (DE 3, 4, 7-12), the remaining ATAC-called elements (+319, +325, +389, +405, +417, +520) included yet uncharacterized elements...”. The authors have forgotten to mention +515.

c. In materials and methods:

- A reference should be added in the following sentence: “...To this purpose, short reads were aligned to the mm10 assembly of the mouse genome using bowtie (ref), with the following parameters: -a -m 1 -n 2 -l 32 -e 3001.”.

- The meaning of FVB should be added in the sentence: “Embryonic forelimbs, mandibular processes, and hearts from wildtype FVB embryos at E11.5 were micro-dissected in cold 1xPBS”.

- The following sentence is unclear: “Cells were resuspended in 10% FCS (in PBS) and formaldehyde (37%) diluted to a final 2% in a total volume of 1ml was added for fixation for 10 min, as previously described.”.

- The following sentence is unclear: “Subtraction maps were directly generated from Cooler balanced HiCmaps using hicCompareMatrices tool (HiCExplorer v3.7.2) with option ‘--operation diff’. HiCExplorer (v3.7.2) was used to determine normalized inter-domain insulation scores and domain boundaries on Hi-C and subtraction maps using default parameters ‘hicFindTADs -t 0.05 -d 0.01 -c fdr’ computing p-values for minimal window length 50000.”.

- Presence of a typo (repetition) in the following sentence: “RNA was reverse transcribed using SuperScript III (Life Technologies) with poly-dT priming according to manufacturer instructions. qPCR was conducted on a LightCycler 480 (Roche) using KAPA SYBR FAST qPCR Master Mix (Kapa Biosystems) for GDD samples, and on a and on a ViiA 7 Real-Time PCR System using PowerTrack SYBR Green Master Mix (Applied Biosystems) for SV-EnhD samples, according to manufacturer instructions.”.

Reviewer #2 (Remarks to the Author):

In this study by Abassah-Oppong the authors dissect the importance of a gene desert adjacent to *Shox2* in the regulation of its expression throughout mouse development. By deleting the entire gene desert the authors show that it contains a variety of enhancer elements controlling *Shox2* expression in different tissues. For example they show that the elements controlling craniofacial expression are fully within the gene desert. In contrast, limb expression contains enhancers within the gene desert and others located likely in the upstream chromatin domain. The data presented here also confirm results from another recent study on *Shox2* regulation by showing that *Shox2* expression in the heart is controlled by the VS-250 region. This study takes those observations further by identifying a small element within the VS-250 region that controls most of this cardiac expression. The authors also show enhancers within the gene desert contribute to *Shox2* expression and function during skeletal morphogenesis. Finally, the study also includes interesting and high-quality Capture HiC data showing how the structure of this regions changes throughout development and report a very interesting formation of a highly interacting domain that appears cardiac-specific

Overall, the data presented are of very high quality and its conclusions are mostly fully supported by their experiments. My only comment concerns the authors interpretation/suggestion that the highly interacting cardiac domain facilitates the formation of a silent/silencing domain. I would ask the authors to remove this from the results section as there is no data to support this interpretation. In fact the authors themselves mention that there is no accumulation of silent chromatin domains.

Formation of a chromatin domain capable of silencing without enrichment of silencing marks would be a very novel concept but as mentioned I don't think there is any evidence of that. In my opinion the suggestion of negative activity to this region should be restricted to the discussion and should be counter-balanced with the observation that its effect may also be neutral or positive.

To me, the observation of this subdomain may be the most novel/interesting part of this study. Although it may fall beyond the scope of this paper understanding its formation and function would be extremely interesting. An easy way to achieve this would be to understand if it falls in the A or B compartment. Can the authors call compartment in their 3.5Mb CHiC region? If that's too small, is there any heart HiC data where this region is also visible? If so, identifying compartments may elucidate the nature of this chromatin domain. It could also be interesting to look at cohesin occupation in these tissues to understand if CTCF may be delimiting this domain-despite the motifs not being in convergent orientation. But as mentioned, these experiments likely fall beyond the goal of this study.

Other minor comments include:

Since the authors show CTCF in ES cells throughout the paper, showing cohesion would also be helpful

Figure 4C and text- Mouse protein symbols should be in all caps.

Figure 4D- I don't think the virtual 4C adds much. But if showing the FL signal should be also shown, alongside the "vs" track for proper comparison.

Figure 4- given the importance of the SV250 region it would be important to show where this region is in the context of the entire locus. Either by showing the entire locus again in Figure 4, or marking the SV250 region in one of the other figures.

In Figure 5a the authors highlight an open region in the human orthologous sequence of the +325-mouse sinus venosus enhancer in both left and right atrium. Could author explain why Shox2 expression is restricted to the SV/RA while the enhancer region is also open in the left atrium?

Reviewer #3 (Remarks to the Author):

The paper by Abassah-Oppong deals with the regulatory impact of enhancers within a large gene desert, flanking the SHOX2 gene. This gene encodes a transcription factor with pleiotropic

expression and multiple roles during development. Using LacZ reporter assays in mouse at E11.5 as a model, the authors can describe at least 20 different enhancers within the TAD in at least one of the tissues analysed within the timepoints E11.5-15.5: one (core) enhancer affecting the cardiac sinus venosus, an array of at least five proximal limb enhancers, as well as enhancers affecting craniofacial and neuronal cell populations, indicating an enormous complexity. It is also likely that this will not be the end of the story and that further enhancers will be revealed in future. Analysis of the 3D chromatin architecture suggested that the enhancers interact with the promotor likely by a tissue-specific chromatin loop.

Altogether, the authors break down a highly complex issue and deserve recognition for it. The paper has been prepared with great care, presents comprehensive data, is discussed very well and all the (145!) references are correctly cited.

Questions and comments:

1. It is interesting and puzzling that the expression from the DE3 interval and not from the DE8 or the DE12 interval showed the strongest effect in heart (Figure 1B; Figure 4E)? On the other hand, expression of these enhancers are found in the sinus venosus. It is also interesting that the cardiac core enhancer which reduces expression by 60% still leads to newborn mice at normal Mendelian frequency.
2. According to data in Figure 5C the SV Enhancer shows a regulatory effect on both *Shox2* and *Rsrc1* genes – what consequences could this have, taking into account the function of *Rsrc1*?
3. Viewed from a higher level point of view, this paper nicely shows as a model the complex spatiotemporal enhancer dynamics in the regulation of gene expression.

Reviewer #4 (Remarks to the Author):

I co-reviewed this manuscript with one of the reviewers who provided the listed reports as part of the Nature Communications initiative to facilitate training in peer review and appropriate recognition for co-reviewers.

REVIEWER COMMENTS

Reviewer #1 (Remarks to the Author):

In this paper, Abassah-Oppong et al., investigate the role of the gene desert flanking the *Shox2* locus in its transcriptional regulation. *Shox2* encode a transcription factor involved in limb, craniofacial, hindbrain and cardiac morphogenesis. The authors first did a screen to identify putative active enhancer elements in the gene desert. The in vivo activity patterns of these putative enhancer elements were then assessed by performing LacZ transgenic reporter assays in mouse embryos. In total, the authors tested 25 individual enhancer elements (DE1 to DE16, +319kb, +325kb, +389kb, +405kb, +417kb, +515kb, +520kb; PEL3 and PLE5; PLE4/DE4 being the same enhancer element). While some of the tested enhancers displayed no staining (i.e. +319kb, +389kb, +405kb, +417kb, +515kb, +520kb enhancers) or no reproducible staining, the authors also identified enhancers with specific subregional activities, predominantly in limb, craniofacial, neuronal, and cardiac cell populations, i.e. in regions known to be dependent on *Shox2* expression.

To evaluate the function of the gene desert flanking *Shox2*, the authors deleted it using a CRISPR/cas9 system in mouse. They found that the gene desert interval is important for *Shox2* transcriptional control in developing limbs, craniofacial compartments, and in the cardiac sinus venosus. Phenotypically, the authors show that deletion of the gene desert leads to pacemaker-related embryonic lethality due to *Shox2* depletion in the cardiac sinus venosus. They also report a requirement of the gene desert for stylopod morphogenesis. The authors then conclude that the *Shox2* gene desert can be considered as a fundamental genomic unit that controls pleiotropic *Shox2* expression.

The paper is well written, the presented experiments are of high quality and convincing. The amount of work involved in this study is substantial and the approach thorough and highly informative.

On the other hand, some data are not completely novel. Among the four limb enhancers identified in the gene desert, one (i.e. hs1262/LHB-A) was already described. Moreover, the heart enhancer called SV enhancer identified in the present study is part of a broader region (241kb) called VS-250. Deletion of this region was shown to be sufficient to deplete *Shox2* in the cardiac sinus venosus (SV), resulting in a hypoplastic SAN and abnormally developed venous valve primordia responsible for embryonic lethality. It should however be noted that an effort was made by the authors to refine the analysis, since the SV-enhancer identified here represent 4kb within the VS-250 region of 241kb. By inactivating the SV-enhancer, the authors also validated the role of this enhancer in regulating *Shox2* expression in SV; inactivation of the SV-enhancer was however not sufficient to recapitulate the phenotype obtained by the VS-250 deletion, indicating additional roles of other enhancers.

Overall, I believe this elegant study will be of great interest to the field, provided that the following revisions are implemented.

Author remark: We thank the Reviewer for the positive appreciation of our work, as well as for the detailed evaluation and the important suggestions. Our study independently identified the *Shox2* gene desert as an essential genomic region regulating pleiotropic *Shox2* expression and embryonic development, and we thank the Reviewer for appreciating our subsequent effort to refine the *Shox2* cardiac enhancer landscape which led to the identification of the +325 SV enhancer that we validated via generation of a mouse enhancer KO model.

The hs1262/LHB-A limb enhancer described in our previous studies (Osterwalder et al., 2018; PMID: 29420474, Rosin et al., 2013; PMID: 23575226) on its own showed no essential requirement for limb-specific *Shox2* expression and only upon combined deletion with hs741 in a sensitized background exhibited a mild effect on proximal limb development (Osterwalder et al., 2018; PMID: 29420474). These results served as an important starting point for our study with one

of the goals being to investigate whether additional limb enhancers in the gene desert would participate in regulating limb-specific *Shox2* expression. In Figure 6, both hs741/PLE1 and hs1262/PLE2 are important positive controls for the 4C experiment and represent essential components of the newly defined *Shox2* proximal limb enhancer element (PLE) repertoire. We also reveal the spatiotemporal activity pattern of the hs741/PLE1 enhancer (Fig. S6), along with the other newly discovered PLEs.

Main points of revision:

1. While the C-HiC profiles across the *Shox2* TAD and flanking regions are largely similar in mouse embryonic forelimbs, mandibles, and hearts at E11.5, the authors report the presence of a cardiac specific high-density contact domain (HCD) spanning ~170kb in the most distal D-dom compartment. In the model shown in Figure 7, the authors proposed that this HCD may influence heart-specific *Shox2* expression by sequestering non-heart enhancers from interacting with the *Shox2* promoter. However, the role of this cardiac specific HCD is unclear. Indeed, from the figure 1A, it seems that the cardiac enhancer +325kb (which is close to DE3) is interacting with *Shox2* not only in the heart but also in the mandible and in the forelimb, then questioning the role of HCD in allowing the specific interactions between the cardiac enhancer +325kb and *Shox2* in the heart. To clarify this point, the authors should perform a 4C experiment in heart, mandible, and forelimb tissues with a viewpoint on the enhancer +325kb.

Author response: We thank the Reviewer for the depth of the comments and thoughtful suggestions. While in the Discussion we hypothesize that “HCDs could act to topologically sequester regulatory regions for modulation of target gene interaction in a tissue-specific manner”, it was not our intent to imply that the HCD would interfere with the interaction of elements outside, but rather those within the domain. To highlight this more clearly, we have now modified the legend of Fig. 7B stating “...high-density contact domain (HCD) that may influence the activity of contained enhancers”. In addition, we also modified the scheme to better indicate the region essential for cardiac *Shox2* expression (VS-250, Van Eif et al., 2020) expected to contain additional SV enhancers (dashed red line).

In any case, we certainly agree with the Reviewer on the interest of the additional exploration of the tissue-specific interactions of the +325 enhancer and therefore we performed virtual 4C (v4C) analysis using the +325 element as viewpoint in forelimb, mandible and heart datasets. Given the high-quality of the novel CHi-C datasets that we have generated from forelimb, mandible and heart, we opted for v4C analysis instead of classical 4C, as the latter does not allow for the removal of duplicates. These results, now presented as new panel C in Figure S5, corroborate the previous observation from *Shox2* promoter-centric v4C analysis that the +325 region interacts with the *Shox2* promoter in embryonic hearts at E11.5 (Figs. 4D and S3B), seemingly stronger than in limb or mandible tissues, as supported by GOTHiC analysis (Figs. S3B, S5C, Table S3). We now refer to these new results in the last part of the following sentence in the main text (line 296):

“This analysis identified a single element located 325kb downstream of *Shox2* (+325) that was able to drive reproducible LacZ reporter expression specifically in the cardiac SV region in a reproducible manner (Fig. 4E, S5B) and showed interaction with the *Shox2* promoter in hearts at E11.5, as indicated by v4C analysis using viewpoints on the *Shox2* promoter and the +325 element itself (Fig. 4D, S3B, S5C).”

It is moreover not clear if the enhancer +325kb is specific for heart tissue, or if it is also active in the craniofacial region or in the limbs? Indeed, in the Figure 4E, the LacZ staining is focused on the heart. The authors should clarify this point as well.

Author response: We thank the Reviewer for pointing this out. While in the initial version we showed the heart-specific activity of the +325kb element in a whole mount embryo in Fig. S5B (top panel), we have now moved this image to Figure 4 (Fig. 4D) to clarify this point. In addition, we have added a close-up of the +325-LacZ transgenic embryo to Fig. S5B, for better visibility of the SV-specific LacZ activity.

2. In the recent study showing that the deletion of VS-250 within the gene desert is sufficient to deplete *Shox2* in the SV (van Eif et al., 2020), the authors also reported the presence of a regulatory element called R4, outside SV-250, that is highly conserved in mammals and able to drive reporter activity in the *Hcn4*⁺/*Shox2*⁺ SAN of E11.5 mouse embryos. Was this enhancer identified by the first screening analysis performed by the authors?

Author response: We thank the Reviewer for bringing this up. The Van Eif et al. 2020 study (PMID: 33040635) presents a human genomic enhancer element termed R4 able to drive LacZ reporter activity in the SV/SAN in mouse embryos at E11.5. We were however not able to find the respective genomic coordinates of R4 in that elegant study and therefore the exact position and length of the R4 element remained unclear. With the goal to nevertheless map the R4 element to the mouse genome based on peak size and location presented in the Van Eif et al. 2020 study, we have re-processed their ATAC-seq data from SAN pacemaker-like cells (SANLPC; n1-n3) and ventricle-like cardiomyocytes (VLCM; n1-n3) (GSE146044). This allowed us to identify the R4-corresponding SANLPC peak based on overlap with the respective highly conserved element located in the genomic region in between the VS-250 interval and the *SHOX2* gene body. Lift-over of the R4 element to the mouse genome subsequently revealed overlap with the DE1 (+184) element identified in our study (see below, Fig. 1 for Reviewers). However, as DE1 was not driving reproducible LacZ reporter activity in the heart/SV (Fig. 1C), but instead in the cranial nerve (JGn), we conclude that the SV activity of the human R4 element must be dependent on human-specific sequence elements. This also would be in agreement with the exclusive requirement of the VS-250 region (not containing R4/DE1) for *Shox2* expression in the SV (Van Eif et al., 2020).

Fig. 1 for Reviewers. Overlap of the DE1 element (mouse liftover, this study) with the presumptive R4 peak observed in human SANLPCs (Van Eif et al., 2020; PMID: 33040635).

At any rate, it would be important for completeness to cite it as a putative identified SV enhancer and to display it, in figure 1A, with the other identified *Shox2* enhancers in the gene desert.

Author response: We thank the Reviewer for this valid suggestion. Based on the mapping described in the above section, we are now showing the genomic location of the R4 element (conserved mouse ortholog) in the locus scheme in Fig. 1A. Both the notion that R4 was identified as a human cardiac SV enhancer by Van Eif et al., 2020, as well as the finding that the conserved core of this element maps to DE1 are now mentioned in the main text (line 155 and 191, respectively).

More generally, it can be helpful to systematically (on all figures) place the already identified enhancers, in addition to the ones identified here, on the gene desert genomic coordinates.

Author response: We thank the Reviewer for this suggestion and have implemented this in all figures now.

3. The authors mention that: “Neither knockout of the hs1262 proximal limb enhancer nor the identification of new limb enhancers (DE6, DE10) located in the gene desert (Fig. 1) was sufficient

to explain the ~50% *Shox2* reduction observed in proximal fore- (FL) and hindlimbs (HL) of GDD/D embryos”; however, the reduction of *Shox2* expression in GDD/*Shox2*Dc mutants, as compared to the reduction of *Shox2* expression resulting from *hs1262* alone is not shown. The role and added value of PLE3-5 enhancers on the transcriptional regulation of *Shox2* as compared to *hs1262* has not been evaluated.

Author response: We thank the Reviewer for this rigor and thoughtful comments. We agree that defining the “reduction of *Shox2* expression in GDd/*Shox2*dc mutants” is a valuable addition allowing to formally link the proximal skeletal phenotype observed in GDd/*Shox2*dc mutants (Fig. 6, S7) to a reduction of *Shox2* dosage. To address this, we performed real-time qPCR from GDd/*Shox2*dc mutant embryonic fore- and hindlimbs at E11.5 along with wildtype (WT) and *Shox2*^{Dc/+} controls. In accordance with the respective proximal limb phenotypes, our qPCR results now reveal that merely 20% and 25% of *Shox2* transcript levels are present in fore- and hindlimbs of GDd/*Shox2*dc embryos, as compared to ~50% in *Shox2*dc/+ controls.

Hs1262 knockout (KO) mice were part of our previous study (Osterwalder et al., 2018; PMID: 29420474) and were assessed in a different mouse strain (pure FVB) background. In addition, revivification (IVF) of the line (sperm stored at Lawrence Berkeley National Laboratory), transfer to UC Calgary (Lab of John Cobb), and breeding into the existing *Prx1*-Cre lines was not feasible within the restricted time-frame of these revisions. However, as shown in our previous study, in absence of both *hs741* and *hs1262* enhancers in *Shox2*d background, *Shox2* is retained at ~35% in hindlimbs (Osterwalder et al., 2018; Data Figure 7f). Hereby, as demonstrated by the individual *hs741* and *hs1262* enhancer KOs, the reduction in *Shox2* expression is mainly a result from lacking the (upstream) *hs741* enhancer located outside of the gene desert: *hs741*KO: 20% reduction of *Shox2* expression; *hs1262* KO: *Shox2* reduction not significant (Osterwalder et al., 2018; Data Figure 2b). In summary, we conclude that additional PLEs (such as PLE3-5) must contribute to phenotypically relevant *Shox2* levels in the limb.

The new qPCR results are now shown in Fig. 6D and the *Shox2* transcript levels relevant for interpretation of the functional contribution of PLEs are now listed in the main text. We have adjusted the respective paragraph as follows (line 396):

“Remarkably, this abolishment of gene desert-mediated *Shox2* regulation in limbs led to a reduction of around 25-30% of *Shox2* transcripts in fore- and hindlimbs of GDd/*Shox2*dc embryos at E11.5, as compared to *Shox2*dc/+ heterozygote controls (Fig. 6D). This reduction surpasses the effect of PLE1 (*hs741*) / PLE2 (*hs1262*) double enhancer loss in *Shox2*-deficient background in hindlimbs (~15% reduction)^{20,28}, suggesting significant functional contributions of PLEs other than PLE2/*hs1262* within the gene desert. In agreement, GDd/*Shox2*dc skeletons revealed more severe shortening of the stylopod than *hs741*/*hs1262* double enhancer knockouts^{20,28}, with an approximate 60% reduction in humerus length and 80% decrease in femur extension in GDd/*Shox2*dc newborn mice (Fig. S7A, B).”

Also, can the authors clarify whether, in their analysis to identify putative *Shox2* active limb enhancers, they found the previously described limb enhancer *hs638*

Author response: *Hs638* is a human element that showed reproducible enhancer-reporter activity in the in the proximal-most domains in the anterior and posterior border zones of limb and lateral plate mesenchyme (see Vista Enhancer Browser). *Hs638* was not among the elements (DEs) identified by our epigenomic analysis (Figs. 1B, S1A), likely due to the restricted activity in the limb mesenchyme. We have now modified panels 1A and 1B (also addressing point 5 below) to indicate the exact position of DEs. To ensure that the limb activity of *hs638* is conserved in the respective mouse orthologous element (*mm2107*), we have in addition performed *Hsp68*-LacZ reporter transgenesis at E11.5 which confirmed a reproducible pattern in n=2/4 embryos. We have included this result in Fig. 1A for better clarity.

4. To estimate the number of developmental enhancers in the gene desert, the authors established a map of stringent enhancer activities by using ChromHMM-filtered H3K27ac signatures from bulk tissues across 66 embryonic and perinatal tissue-stage combinations, from ENCODE. The time

points included in the analysis were E11.5 -15.5. However, the validity of such an approach, even for initial screening, is questionable, given the risk of missing essential enhancers. Indeed, neither the SV enhancer +325kb, nor the limb enhancers PLE3 and PLE5 were identified using this method. The authors moreover mention that “the previously validated 5 and hs1262 limb enhancers were not among the predictions as elevated H3K27ac marks are present at these enhancers already at earlier stages.” However, as shown in the vista browser and in figure 1A, hs741 and hs1262 are still active at E11.5 (a stage included in the analysis). Shouldn't they then have been identified by the ChromHMM-filtered H3K27ac signature analysis? It would have been more appropriate to dissect mouse tissues at appropriate stages, where *Shox2* is known to be active, and perform H3K27ac ChIPseq +/-, ATACseq +/-, and chromatin conformation capture (C-HiC, 4C-seq). This is what the authors finally did, but only for limbs. It is regrettable that a comparable methodical approach was not taken for the craniofacial and cardiovascular domains.

Author response: We thank the Reviewer for the depth of these comments. Given the frequently complex pleiotropic expression pattern of *Shox2*, as well as the spatio-temporal dynamics of each expression domain, we decided to use the ENCODE resource as it currently represents the most comprehensive collection of histone marks across a large number of *Shox2* expressing embryonic tissues at relevant developmental timepoints. Dissecting a comparably large number of (*Shox2*-expressing) tissues in individual embryos and across a similar range of timepoints would have been highly time-consuming, with the risk to affect dissection accuracy and sample quality, especially for *Shox2*-expressing compartments limited in cell number (e.g. sinus venosus, cranial nerve, craniofacial subregions). Alternatively, sorting of *Shox2*-expressing cells would have required the generation of a *Shox2*-fluorescent reporter mouse line.

We performed however additional analysis focused on limb, craniofacial, heart, midbrain and hindbrain datasets to provide a list of more threshold-relaxed (lower-probability) and temporally extended predictions within the main *Shox2* expressing tissues. To this purpose, we applied different H3K27ac-thresholds (qval=0.05, 0.001, 0.00001) and included the more recent E10.5 ENCODE datasets to extend predictions, which led to a larger numbers of candidate regions also including hs1262, but not hs741. These predictions are now presented in **Table S1** (*Sheets 4 and 5*) and referenced in the main text as follows (line 169): “Reducing stringency of H3K27ac-thresholds and including E10.5 profiles however extended the number of predicted enhancer elements within the *Shox2* TAD significantly (Table S1).”

As stated in the Discussion (line 451), while H3K27ac signatures are considered one of the most valuable genome-wide predictors of enhancer activity, there are limitations, and our study provides further evidence for discrepancies of H3K27ac-predictions and reporter activities at the tissue level (occurrence of false positives/false negatives). For example, the temporal H3K27ac profiles of hs741 (PLE1) and hs1262 (PLE2) elements differ from the temporal LacZ reporter profile in limbs which persist until at least E13.5 (see Figure 2 for Reviewers below, Fig. 1A, S6B and Rosin et al., 2013, PMID: 23575226). For better clarity we have modified the results text as follows (line 166): “The previously validated hs741 and hs1262 limb enhancers were not among the stringent predictions across timepoints as H3K27ac levels at these enhancers are progressively reduced after E10.5⁵⁷, despite strong LacZ reporter signal in the proximal limb at subsequent stages (Fig. 1A)^{20,58}.”

In addition, for better comparison of temporal H3K27ac profiles among all PLEs we have added a new panel (A) in Figure S6 now showing detailed H3K27ac profiles in limb buds at E10.5 and in proximal and distal limb compartments at E12.5 for all PLEs (analogous to the locus-wide panel of Fig. 6A).

Fig. 2 for Reviewers. H3K27ac profiles at selected *Shox2*-associated enhancers in mouse embryonic limbs at E10.5 to E15.5 from ENCODE (PMID: 32728249; <https://screen.encodeproject.org>). Dashed line indicates stages used in the stringent epigenomic profiling analysis for enhancer prediction (Fig. 1A). hs741, hs1262: enhancers driving LacZ reporter activity in the limb from E10.5-E13.5 (Fig. S6B, Rosin et al., 2013). DE4: enhancer equivalent to PLE4 driving LacZ reporter activity in the limb from E11.5-E13.5 (Fig. S6). DE15: enhancer driving craniofacial activity but no limb activity at E11.5 (Fig. 1C, 3F).

5. The presented heatmap in Figure 1B is misleading as it suggests that the gene desert enhancers DE1 to DE16 completely cover the gene desert, with each enhancer element being of the same size. However, this is not the case. For clarity, it would be helpful to display boxes on the genomic coordinate of the gene desert presented in Figure 1A, indicating the location and size of each enhancer element.

Author response: We thank the Reviewer for this excellent suggestion. We have modified Figure 1 to display the exact position of elements DE1-16 within the gene desert (Fig. 1A). While the scale in Fig. 1A is too large to visualize the extension of DE elements, Fig. 4D depicts the size of DE3-DE12 at higher resolution later in the study. The exact basepair size of all DE elements is listed in Table S2.

6. In their analysis using the LacZ transgenic reporter on DE1 to DE16, the authors noted reproducible activity in 12 out of 16 cases tested. Are these 12 cases those depicted in figure 1C? If so, does this imply that DE4/PLE4 (not displayed in Figure 1C at E11.5, but visible in Figure 6C at E12.5) does not exhibit a repeatable pattern? The authors should clarify this issue and specifically mention which enhancer(s) among the DE1-DE16 do not result in a repeatable pattern.

Author response: We thank this Reviewer for this rigor and valuable suggestion. It is correct that originally the 12 out of 16 cases with reproducible activities at E11.5 were represented in Figure 1C. In the course of these revisions and thanks to this Reviewer's input we were able to make two important updates:

1. Re-analysis of all transgenic embryos identified DE2 as additional enhancer element driving reproducible LacZ reporter activity in the 3rd pharyngeal arch. DE2 is now included in Fig. 1C as a positive element. For all enhancers identified, images of LacZ transgenic embryos (including replicates) have been deposited at the Vista Enhancer Browser (<https://enhancer.lbl.gov>), including close-ups on the heart.
2. In the submitted manuscript version we did not include DE4 (Fig. 3A for Reviewers below) as element driving (limb) enhancer activity in Fig. 1C as the reproducibility of the pattern driven by the DE4 element remained unclear. Additional analysis of DE4-LacZ transgenic reporter embryos in the course of these revisions revealed a second replicate embryo with overlapping activity in the proximal limb at E11.5 (see Fig. 3B for Reviewers below), highly similar to the patterns observed in PLE4-hsp68-LacZ transgenic embryos at E11.5 (Fig. S6B). In addition, as part of an experiment of a follow-up study, we have confirmed that DE4 is active at later stages and drives a PLE4-like pattern at E14.5 (compare PLE4 limb

activity at E13.5 in Fig.S6B to DE4 activity at E14.5 in Fig. 3C for Reviewers below). Altogether, these results allow us to conclude that the activities driven by DE4 and PLE4 (differing in a PLE4 extension of 486bp) are equivalent and reproducible, and consequently DE4 is now included as a limb enhancer in Fig. 1C.

These additions leading to n=14/16 enhancers identified in Fig. 1C are now also updated in the main text (line 177) and, as requested by this Reviewer, we now also specifically mention the elements without reproducible LacZ reporter activity in the main text (line 200): “DE3 (n=7) and DE13 (n=5) were the only elements not showing any reproducible activities in transgenic LacZ reporter embryos at E11.5”.

Fig. 3 for Reviewers. (A) Extension of PLE4 and DE4 elements along with H3K27ac and H3K27me3 profiles in proximal and distal limbs at E12.5 (see also Fig. 6A and new Fig. S6A). (B) Left: Forelimbs from two distinct DE4-Hsp68-LacZ transgenic reporter embryos showing overlapping LacZ reporter signal at E11.5 (n=2/5). Right: PLE4-Hsp68-LacZ embryo (n=2/2) showing near-identical proximal limb activity at E11.5 (embryo shown in Fig. S6B). (C) DE4 activity in mouse embryos at E14.5 driven by a DE4-bG-LacZ cassette integrated at the neutral H11 locus using enSERT (Kvon et al., 2020, PMID: 32169219). bG: beta-Globin minimal promoter.

7. The authors mention “Other prominent 4C-seq peaks in the gene desert co-localized with either previously validated enhancer elements with non-limb activities at E11.5 (DE1, 4, 6, 9, 15) or non-validated elements with proximal limb-specific H3K27ac enrichment (+237kb and 346 +568kb)”. The LacZ staining for DE4 at E11.5 is however not shown; it is not clear if DE4 is among the DEs that do not show a reproducible pattern at E11.5 (see also above).

Author response: In light of the corrected interpretation for DE4 (see above, point 6), we have now included the LacZ image of DE4 in Fig.1C as an additional active enhancer. The corresponding main text has been adjusted in the following paragraph including also aspects of points 8 and 9 below (line 357):

“Other prominent 4C-seq peaks in the gene desert co-localized with either previously validated enhancer elements with non-limb activities at E11.5 (DE1, 6, 9, 15) or non-validated elements with proximal limb-specific H3K27ac enrichment (+237kb and +568kb) (Fig. 6B, C). Epigenomic profiles further revealed that the *Shox2*-interacting DE4 (+407) element showing restricted LacZ activity in the proximal limb at E11.5 (n=2/5) was unique in its H3K27ac pattern initiated past E10.5, while other (candidate) proximal limb enhancers (PLEs) showed H3K27ac enrichment present already at E10.5 (Fig. 6A, S6A). Therefore, we decided to analyze the spatiotemporal activities of newly identified (+237kb, +568kb) and seemingly temporally dynamic (DE4, +407) (candidate) limb enhancer regions using stable transgenic LacZ reporter mouse lines.”

8. The authors state that: “Open chromatin and H3K27ac profiles further indicated that the *Shox2*-interacting DE4 (+407) element was unique in its H3K27ac pattern (initiated at E11.5), while other proximal limb (candidate) enhancers showed activity marks already at E10.5.” They should refer to the figure where they show this. Indeed, the authors mention that they “reprocessed previously published ChIP-seq datasets from dissected proximal and distal limbs at E12”; but they never mentioned that they analyzed open chromatin and H3K27ac profiles at earlier stages.

Author response: We thank the Reviewer for this valuable observation and suggestion. We were originally referring to the locus-wide map in Fig. 6A for the tracks at E10.5, but we acknowledge that the individual enhancer signatures appear rather small and therefore inspection of the exact profiles is challenging. Using the same track setup, we have now added an extra panel (Fig. S6A) showing the profiles for each PLE (1-5) at higher resolution. We refer to these figures now in the respective sentence as requested (line 360):

“Epigenomic profiles further revealed that the *Shox2*-interacting DE4 (+407) element showing restricted LacZ activity in the proximal limb at E11.5 (n=2/5) was unique in its H3K27ac pattern initiated past E10.5, while other (candidate) proximal limb enhancers (PLEs) showed H3K27ac enrichment present already at E10.5 (Fig. 6A, S6A).”

9. The authors state that: “Open chromatin and H3K27ac profiles further indicated that the *Shox2*-interacting DE4 (+407) element was unique in its H3K27ac pattern (initiated at E11.5), while other proximal limb (candidate) enhancers showed activity marks already at E10.5.”. Then, it is mentioned that “...PLE4 (+407) drove reporter activity already at E10.5 in a more widespread pattern leaking into distal forelimbs at later stages, in line with H3K27ac enrichment in distal forelimbs (Figs. 6A, S6A).”

The authors may explain this discrepancy between the time when H3K27ac starts to be observed (E11.5) and the time at which the reporter is observed (E10.5) in the LacZ transgenic reporter assay.

Author response: We thank the Reviewer for this critical observation and rigor. Unfortunately, the PLE4 embryo image (at E10.5) has been misinterpreted and upon careful re-examination we realized that the LacZ staining is actually located behind the limb bud (see Fig. S6B). This allows us to conclude that, in accordance with absence of H3K27ac at E10.5 (Fig. S6A), there is no PLE4 activity at E10.5. We would like to apologize for this oversight and have now also added dashed lines circling the forelimb buds at E10.5 for better visualization (Fig. S6B). The respective sentence has been modified as follows (line 380):

“Similarly, PLE4/DE4 (+407) activity emerged at E11.5 in the proximal-posterior (see also Fig. 1C) but extended in a more widespread fashion into distal limbs at later stages, in line with elevated H3K27ac in distal forelimbs at E12.5 (Figs. 6A, S6A, S6B).”

Minor points:

a. The authors state that “... .Despite such critical roles, the functional requirement of only few gene deserts have been studied in detail, including the investigation of chromatin topology and functional enhancer landscapes in the TADs of other developmental key regulators such as *Sox9*, *Shh* or *Fgf8*^{10–12}.”. For the sake of completeness, an additional recent study could be cited here that identified functional long-range enhancers in a gene desert involved in the regulation of the key craniofacial regulator *Hoxa2* (Kessler et al., Nat Comm 2023).

Author response: We thank the Reviewer for this valuable suggestion and agree that the Kessler et al. 2023 (PMID: 37277355) study represents an ideal article to be cited in this position. We have included this reference in the main text as follows (line 85):

“Despite such critical roles, the functional requirement of only few gene deserts have been studied in detail, including the investigation of chromatin topology and functional enhancer landscapes in the TADs of other developmental key transcription factors (TFs), such as *Sox9* and *Hoxa2*^{10,11}, or signaling ligands, such as *Shh* or *Fgf8*^{12,13}.”

b. In the following sentence: “While a subset of these candidate SAN enhancer elements overlapped DEs validated for non-cardiac activities (DE 3, 4, 7-12), the remaining ATAC-called elements (+319, +325, +389, +405, +417, +520) included yet uncharacterized elements...”. The authors have forgotten to mention +515.

Author response: We thank the Reviewer for this observation and apologize for the oversight. "+515" is now included in the list.

c. In materials and methods:

- A reference should be added in the following sentence: "...To this purpose, short reads were aligned to the mm10 assembly of the mouse genome using bowtie (ref), with the following parameters: -a -m 1 -n 2 -l 32 -e 3001."

Author response: We apologize for the missing reference – the Bowtie reference is now added (line 621).

- The meaning of FVB should be added in the sentence: "Embryonic forelimbs, mandibular processes, and hearts from wildtype FVB embryos at E11.5 were micro-dissected in cold 1xPBS".

Author response: We have specified this now by stating "FVB strain mouse embryos" (line 637).

- The following sentence is unclear: "Cells were resuspended in 10% FCS (in PBS) and formaldehyde (37%) diluted to a final 2% in a total volume of 1ml was added for fixation for 10 min, as previously described."

Author response: the sentence is now corrected to the following (line 639): "Cells were resuspended in 10% FCS (in PBS) and 1ml of formaldehyde (37% in H₂O, Merck) diluted to a final 2% was added for fixation for 10 min, as previously described⁶²."

- The following sentence is unclear: "Subtraction maps were directly generated from Cooler balanced HiCmaps using hicCompareMatrices tool (HiCExplorer v3.7.2) with option '--operation diff'. HiCExplorer (v3.7.2) was used to determine normalized inter-domain insulation scores and domain boundaries on Hi-C and subtraction maps using default parameters 'hicFindTADs -t 0.05 -d 0.01 -c fdr' computing p-values for minimal window length 50000."

Author response: these are two subsequent sentences and we have slightly modified them for more clarity (line 670): "Subtraction maps were directly generated from Cooler balanced HiC maps using the hicCompareMatrices tool (HiCExplorer v3.7.2) with option '--operation diff'. HiCExplorer¹²⁶ (v3.7.2) was used to determine normalized inter-domain insulation scores and domain boundaries on Hi-C and subtraction maps using default parameters 'hicFindTADs -t 0.05 -d 0.01 -c fdr' computing p-values for a minimal window length of 50000."

- Presence of a typo (repetition) in the following sentence: "RNA was reverse transcribed using SuperScript III (Life Technologies) with poly-dT priming according to manufacturer instructions. qPCR was conducted on a LightCycler 480 (Roche) using KAPA SYBR FAST qPCR Master Mix (Kapa Biosystems) for GDD samples, and on a and on a ViiA 7 Real-Time PCR System using PowerTrack SYBR Green Master Mix (Applied Biosystems) for SV-EnhD samples, according to manufacturer instructions."

Author response: we thank the Reviewer for this observation and have corrected the typo (line 727):

"...using KAPA SYBR FAST qPCR Master Mix (Kapa Biosystems) for GD^d samples, and on a ViiA 7 Real-Time PCR System using PowerTrack SYBR Green Master Mix..."

Reviewer #2 (Remarks to the Author):

In this study by Abassah-Oppong the authors dissect the importance of a gene desert adjacent to Shox2 in the regulation of its expression throughout mouse development. By deleting the entire gene desert the authors show that it contains a variety of enhancer elements controlling Shox2 expression in different tissues. For example they show that the elements controlling craniofacial expression are fully within the gene desert. In contrast, limb expression contains enhancers within the gene desert and others located likely in the upstream chromatin domain. The data presented here also confirm results from another recent study on Shox2 regulation by showing that Shox2 expression in the heart is controlled by the VS-250 region. This study takes those observations further by identifying a small element within the VS-250 region that controls most of this cardiac expression. The authors also show enhancers within the gene desert contribute to Shox2 expression and function during skeletal morphogenesis. Finally, the study also includes interesting and high-quality Capture HiC data showing how the structure of this regions changes throughout development and report a very interesting formation of a highly interacting domain that appears cardiac-specific

Overall, the data presented are of very high quality and its conclusions are mostly fully supported by their experiments.

Author remark: We are delighted about this overall very positive feedback from Reviewer #2 and are grateful for the inspiring and thoughtful comments.

1. My only comment concerns the authors interpretation/suggestion that the highly interacting cardiac domain facilitates the formation of a silent/silencing domain. I would ask the authors to remove this from the results section as there is no data to support this interpretation. In fact the authors themselves mention that there is no accumulation of silent chromatin domains. Formation of a chromatin domain capable of silencing without enrichment of silencing marks would be a very novel concept but as mention I don't think there is any evidence of that. In my opinion the suggestion of negative activity to this region should be restricted to the discussion and should be counter-balanced with the observation that its effect may also be neutral or positive.

Author response: We thank the Reviewer for this key point and indeed it was not our intention to claim evidence for the existence of a chromatin domain capable of silencing without enrichment of silencing marks. In the revised manuscript, we have made sure that we do not directly make such a claim nor use of the term "silencing" in the results section. We have modified the following paragraph stating the possibility of a repressive function as our initial hypothesis that then however is rendered unlikely by the observation of the absence of repressive compartment type (addressing point 2 below) or histone modifications (line 221):

"While this high-density contact domain (HCD) contained the majority of the previously identified (non-cardiac) gene desert enhancers (DE5-12), subtraction analysis further corroborated increased chromatin contacts across the HCD and domain insulation specifically in cardiac tissue as opposed to limb or mandibular tissue, potentially indicating a repressive function in heart cells due to condensed chromatin state (Fig. 2A-C, S2A-C). However, no region-specific accumulation of repressive histone marks (H3K27me3 or H3K9me3, ENCODE) was observed in whole heart samples (Figs. S2B, S3A, B)."

As this Reviewer suggests, in the Discussion we have also counter-balanced our notion about a potentially negative effect of high-density contact domains on transcription with the possibility of positive or neutral effects (line 511): "While such domains might have inhibiting or augmenting impact on tissue-specific regulatory interactions, a neutral effect may also be possible."

2. To me, the observation of this subdomain may be the most novel/interesting part of this study. Although it may fall beyond the scope of this paper understanding its formation and function would be extremely interesting. An easy way to achieve this would be to understand if it falls in the A or B compartment. Can the authors call compartment in their 3.5Mb CHiC region? If that's too small,

is there any heart HiC data where this region is also visible? If so, identifying compartments may elucidate the nature of this chromatin domain. It could also be interesting to look at cohesin occupation in these tissues to understand if CTCF may be delimiting this domain-despite the motifs not being in convergent orientation. But as mentioned, these experiments likely fall beyond the goal of this study.

Author response: Indeed, we also consider the high-density contact subdomain (HCD) we identify as highly interesting. We appreciate and share the Reviewer's vision that studying the mechanisms of formation and function of the high-density contact subdomain (HCD) is beyond the scope of this manuscript, but we do intend to address these aspects in follow-up studies. To answer the Reviewer's suggestion, there are, to the best of our knowledge, no HiC datasets (at comparable depth) nor ChIP-seq data for CTCF or cohesin subunits available from mouse embryonic heart tissue at comparable stages.

There are however CTCF ChIP-seq tracks from mouse hearts at newborn stage (p0) available from the ENCODE repository, and as an approximation we have now added these profiles to the CTCF tracks in Figures 2B and S2B. There is some cardiac CTCF enrichment present at the HCD border regions, suggesting CTCF to be occupying these regions also in the developing heart. To further explore the presence of cohesin at these sites bound by CTCF in mESCs, we have reprocessed available datasets for cohesin subunits SMC1A and RAD21 from mESCs (Justice et al., 2020; PMID: 32294452) for comparison (now shown in Fig. S2B). In contrast to significant CTCF enrichment in the HCD-flanking regions, no significant cohesin is observed, although minimal levels are present.

As suggested by this Reviewer, we attempted to call A/B compartments from our C-HiC datasets (limb, mandible, heart) to further explore the nature of this domain. Using HiCExplorer (Wolff et al. 2018, PMID: 29901812) and employing the "Lieberman method" (Miura et al. 2018, PMID: 30218370, see Fig. 4 for Reviewers below, panel A, taken from Miura et al. 2018) we however found that our datasets do not provide sufficient region coverage and long-range interaction depth to determine how to group regions together. To evaluate the method, we used whole chromosome 3, 100Mb and 50Mb region input from the distal limb dataset of Rodriguez-Carballo et al., 2017 (PMID: 29273679). This revealed that Hi-C profiles of regions larger than 50Mb are required to reliably call A/B compartments based on the Eigenvector of the 1st principal component (PCA1) (see Fig. 4 for Reviewers below, panel B). In contrast, our C-HiC probe interval (Fig. S2) is spanning "only" 3.5Mb. In addition, given the limited size of the HCD (~170kb), A/B compartment calling would require Hi-C datasets with an even higher resolution, as classical A/B compartment calling is performed on 100kb to 500kb bins (Miura et al. 2018). We were indeed unable to call A/B compartments on the Rodriguez-Carballo et al. (2017) dataset at 40kb resolution due to the sparsity of data and had to use 400kb resolution instead.

Fig. 4 for Reviewers. (A) Scheme of a current state-of-the-art method to call A/B compartments from Hi-C datasets (taken from Figure 1 of Miura et al. 2018). (B) Successful A/B compartment calling based on the sign of the Eigenvector of PC1 requires Hi-C read counts from an extended genomic interval. Hi-C contact, observed/expected and Pearson correlation matrices at 400kb resolution were generated for A/B compartment calling using pairs falling in the following chromatin interval: whole chromosome 3 (“whole chromosome”), chr3:0-100Mb (“100Mb”) and chr3:30-80Mb (“50Mb”). Results are plotted on the chr3:50-70Mb region. Successful compartment calling was only achieved for whole chromosome and 100Mb sized input regions, but not for the 50Mb input region.

Other minor comments include:

Since the authors show CTCF in ES cells throughout the paper, showing cohesion would also be helpful

Author response: We agree with the Reviewer and as already indicated in point 2 above, we are showing ChIP-seq datasets from mouse ESCs now for the cohesin subunits SMC1A and RAD21 from a recent study (Justice et al., 2020; PMID: 32294452).

Figure 4C and text- Mouse protein symbols should be in all caps.

Author response: The mouse protein IDs have been adjusted to “all caps” format in Fig. 4C and Fig. S4D and the corresponding figure legends and main text.

Figure 4D- I don't think the virtual 4C adds much. But if showing the FL signal should be also shown, alongside the "vs" track for prppoer comparison.

Author response: The virtual 4C forelimb (FL) signals (gray/green) have been added to Fig. 4D, analogous to the heart (HT) track (gray/red).

Figure 4- given the importance of the SV250 region it would be important to show where this region is in the context of the entire locus. Either by showing the entire locus again in Figure 4, or marking the SV250 region in one of the other figures.

Author response: The extension of the VS-250 region is now shown in Fig. 6A and Fig. S5C.

In Figure 5a the authors highlight an open region in the human orthologous sequence of the +325-mouse sinus venosus enhancer in both left and right atrium. Could author explain why Shox2 expression is restricted to the SV/RA while the enhancer region is also open in the left atrium?

Author response: We thank the Reviewer for this remark. As observed in our and other studies, as pointed out in the Discussion and in the response to Reviewer #1 (point 4), H3K27ac signatures do not always accurately predict the *in vivo* enhancer activities defined by transgenic reporter assays. While H3K27ac signals can identify *in vivo* enhancer activities across restricted temporal windows (Nord et al., 2013; PMID: 24360275), in many cases H3K27ac signals of a putative enhancer region are detected in multiple tissues, while enhancer activity is detected in only a single tissue type or subregion (compare Figure 1B and C). False positives (or negatives) have been observed particularly in dedicated studies comparing H3K27ac predictions to enhancer-reporter activities in mouse embryos (e.g., Mannion et al., 2022; bioRxiv; doi: <https://doi.org/10.1101/2022.05.29.493901>, Monti et al., 2017).

From these observations we (and others) conclude that H3K27ac alone cannot be used as an exclusive enhancer mark and that in combination with other, e.g. repressive chromatin marks, H3K27ac could be present as part of a poised (yet inactive) enhancer state (Cruz-Molina et al., 2017; PMID: 28285903) in certain related tissues. Such a scenario could explain the presence of H3K27ac at the +325-mouse SV enhancer interval in LA tissue, in addition to RA/SV tissue exhibiting likely active enhancer function.

Reviewer #3 (Remarks to the Author):

The paper by Abassah-Oppong deals with the regulatory impact of enhancers within a large gene desert, flanking the SHOX2 gene. This gene encodes a transcription factor with pleiotropic expression and multiple roles during development. Using LacZ reporter assays in mouse at E11.5 as a model, the authors can describe at least 20 different enhancers within the TAD in at least one of the tissues analysed within the timepoints E11.5-15.5: one (core) enhancer affecting the cardiac sinus venosus, an array of at least five proximal limb enhancers, as well as enhancers affecting craniofacial and neuronal cell populations, indicating an enormous complexity. It is also likely that this will not be the end of the story and that further enhancers will be revealed in future. Analysis of the 3D chromatin architecture suggested that the enhancers interact with the promotor likely by a tissue-specific chromatin loop.

Altogether, the authors break down a highly complex issue and deserve recognition for it. The paper has been prepared with great care, presents comprehensive data, is discussed very well and all the (145!) references are correctly cited.

Author remark: We would like to thank the Reviewer for this very appreciative comment and are delighted about the positive reception and detailed evaluation.

Questions and comments:

1. It is interesting and puzzling that the expression from the DE3 interval and not from the DE8 or the DE12 interval showed the strongest effect in heart (Figure 1B; Figure 4E)? On the other hand, expression of these enhancers are found in the sinus venosus.

Author response: We thank the Reviewer for this thoughtful comment. As our epigenomic prediction approach is based on ENCODE H3K27ac profiles from whole mount tissues predominantly at E11.5-E15.5 (Fig. 1B, S1A), enrichment scores depend on the ratio of cells with a signal vs. cells without. Given the frequent activity of heart enhancers across multiple cardiac compartments, we reasoned originally that SV activities might be coupled to broader cardiac activities, which would appear in the profiling matrix. It is however plausible that signals from SV-restricted enhancers become too diluted in a whole heart sample compared to signals of enhancers that are more broadly active in the heart. In accordance with this hypothesis, the SV-restricted enhancer region (+325) itself was not picked up by the initial H3K27ac-based epigenomic prediction approach (see Fig. 1B, 1C), and also not with relaxed parameters now added in Table S1(sheet 4).

Based on our stringent enhancer predictions DE8 and DE12 indeed stood out as most promising candidates for cardiac enhancers with continuous activity profiles. However, while lacking H3K27me3 in the heart from E10.5-E15.5 (ENCODE) which would indicate a poised state (Fig. S3A), these elements did not drive reproducible cardiac *in vivo* enhancer-reporter activities at E11.5 (Fig. 1C, Vista Enhancer Browser). As at both elements cardiac H3K27ac shows strongest enrichment at progressed stages (E13.5-E14.5, Fig. S1), we decided to evaluate the possibility of later heart enhancer activity in the course of these revisions (Fig. 5 for Reviewers below). To achieve improved efficiency and higher specificity of transgenic results we used the H11 site-directed transgenic reporter framework in mouse embryos (Kvon et al., 2020; PMID: 32169219) that we already utilized for validation of cardiac-specific (SV) activity driven by the +325 enhancer (see Fig. 4E and Fig. 5 for Reviewers below). However, both elements were unable to drive reproducible transgenic LacZ reporter expression in mouse embryos at E13.5, while we obtained LacZ signal in an embryo with random integration (corroborating the functionality of the cassette). As highlighted in the Discussion and recent studies recent studies (e.g. Mannion et al., 2022; bioRxiv; doi: <https://doi.org/10.1101/2022.05.29.493901>, Monti et al., 2017), H3K27ac signatures show limited accuracy in prediction of temporal and tissue-specific enhancer activity *in vivo*, leading to a fraction of false-positive and false-negative predictions. We cannot rule out, however, cardiac activity of DE8 or DE12 elements at even later stages.

Fig. 5 for Reviewers. Assessment of cardiac enhancer activities of DE8 and DE12 core elements at cardiac maturation stages. **(A)** Open chromatin profiles (red) from mouse embryonic hearts at E11.5 along with H3K27ac profiles (yellow) from mouse embryonic hearts at E10.5-E15.5 for DE8, DE12 and +325 (SV enhancer) elements. **(B)** *In vivo* enhancer validation using H11 site-directed reporter transgenesis for increased accuracy and specificity (Kvon et al., 2020; PMID: 32169219) based on a construct involving a beta-Globin (bG) minimal promoter in front of a LacZ reporter (bG-LacZ). While this approach identified specific SV enhancer activity of the +325 element at E11.5 (as shown in Figs. 4E and S5B in the manuscript), no LacZ signal was detected in DE8 or DE12 transgenic reporter embryos at E13.5.

It is also interesting that the cardiac core enhancer which reduces expression by 60% still leads to newborn mice at normal Mendelian frequency.

Author response: We thank the Reviewer for this interesting remark. We agree that this is an interesting finding. Embryos lacking the gene desert (our study) or the SV-250 interval (Van Eif et al., 2020; PMID: 33040635) display complete loss of *Shox2* expression in the heart resulting in embryonic lethality around E12. Instead, embryos with a homozygous deletion of the +325 SV enhancer located within the gene desert/SV-250 interval (our study), retain 40% of *Shox2* expression in the SV (Fig. 5C) and are born without phenotypic abnormalities.

As listed in the Discussion (line 473), a similar observation has also been made for the *Isl1* SAN (ISE) enhancer, which shows a similar embryonic activity profile with reduced target gene expression, but without exhibiting embryonic lethality in a homozygous condition (Galang et al.; PMID: 33044128). It is therefore plausible, that given the severe consequences of losing the expression of TFs essential for the cardiac conduction system and embryonic viability, such as *Shox2* and *Isl1*, there is a significant level of transcriptional robustness built in that allows to compensate the disruption/loss of at least one enhancer. As a major hallmark of a robust system, TFs are frequently still able to exert their crucial functions at significantly reduced gene dosage, indicated by the circumstance that heterozygosity for most TFs, including *Shox2*, in the mouse do not result in developmental phenotypes. In accordance, it has been re-inforced in a recent study that buffering of certain developmental genes can lead to robust, nonlinear dosage-to-phenotype relationships (Naqvi et al., 2023; PMID: 37024583), while others show higher or linear sensitivity. While accurate *SHOX/Shox2* dosage appears to be essential for normal SAN function (Li et al., 2011; PMID: 21454626), minor differences in sensitivity could also be related to the mouse strain background used.

2. According to data in Figure 5C the SV Enhancer shows a regulatory effect on both *Shox2* and *Rsrc1* genes – what consequences could this have, taking into account the function of *Rsrc1*?

Author response: We thank the Reviewer for this question. While downregulation of *Shox2* in homozygous SV enhancer (+325) KO embryos compared to wildtype controls at E10.5 is highly significant, we are not able to conclude that there is a regulatory effect on *Rsrc1* as the p-value (hom vs. wt control) remains non-significant ($P=0.5384$) in the two-tailed, unpaired t-test. We agree however that a minor downregulation could be possible, as indicated by the trend in heterozygous and homozygous knockouts. *RSRC1* is described to play roles in mRNA splicing and (neuronal) transcript regulation in patient fibroblasts and differentiated neural progenitor cells (Perez et al., 2018; PMID: 29522154). While *Rsrc1* is expressed in various tissues with abundant transcript levels in the testis and brain, *Rsrc1* gene KO in mice did not reveal phenotypic abnormalities in fertility, sperm count or motility (Zhang et al., 2023; PMID: 37095490). Altogether, we thus conclude that it is unlikely to detect phenotypic differences from putative minor *Rsrc1* dosage differences.

3. Viewed from a higher level point of view, this paper nicely shows as a model the complex spatiotemporal enhancer dynamics in the regulation of gene expression.

Author remark: As previously mentioned, we very much appreciate the Reviewer's positive evaluation of our study and the insights provided by our fine dissection of the *Shox2* regulatory landscape.

Reviewer #4 (Remarks to the Author):

I co-reviewed this manuscript with one of the reviewers who provided the listed reports as part of the Nature Communications initiative to facilitate training in peer review and appropriate recognition for co-reviewers.

** See Nature Portfolio's author and referees' website at www.nature.com/authors for information about policies, services and author benefits.

This email has been sent through the Springer Nature Tracking System NY-610A-NPG&MTS

Confidentiality Statement:

This e-mail is confidential and subject to copyright. Any unauthorised use or disclosure of its contents is prohibited. If you have received this email in error please notify our Manuscript Tracking System Helpdesk team at <http://platformsupport.nature.com>.

Details of the confidentiality and pre-publicity policy may be found here <http://www.nature.com/authors/policies/confidentiality.html>

Privacy Policy | Update Profile

Figure Modifications included in these revisions:

Main Figures:

Figure 1:

- Panel 1A: Lines have been added to indicate the position of the different DE elements defined in Fig. 1B. The position of the R4 element (Van Eif et al., 2020; PMID: 33040635) is now indicated (Reviewer1, #1). A representative mm2107-LacZ embryo has been added to indicate the reproducible and conserved pattern of the mm2107/hs638 mouse/human elements which are listed in the Vista Enhancer Browser (Reviewer1, #3).
- Panel 1B: A higher quality image of an intact and representative *Shox2*-LacZ embryo at E11.5 has been added to replace the previous image, including also the visualization of *Shox2* activity in the forelimb.
- Panel 1C: Representative transgenic embryos showing reproducible LacZ transgenic reporter activity driven by DE2 (identified after re-inspection of subregional activities) and DE4 (Reviewer1, #6) have been added. For DE8: correction from n=7/9 to n=6/9 DRG positive after re-analysis.

Figure 2:

- Panel 2B: Related to comment #2 of Reviewer 2, CTCF signals from mouse heart at P0 (ENCODE) (shown in orange) were added to overlap the present CTCF profiles from mESCs (in gray). In addition, the locus scheme was updated with additional Vista elements (hs1413, hs741, hs636, hs1262, hs1251, hs638). DE2 and DE4 are marked blue (as included in Fig. 1C).

Figure 3:

- Panel 3A: locus scheme updated with additional Vista elements (hs1413, hs741, hs636, hs1262, hs1251, hs638). DE2 and DE4 are marked blue (as included in Fig. 1C).

Figure 4:

- Panel 4B: embryo cartoon is shown from the right side now for better comparison to the position of hearts shown in panels 4A, C and E.
- Panel 4C: Protein IDs are now in upper case.
- Panel 4D: Corresponding v4C tracks have been added for forelimb (Reviewer2)
- Panel 4E: +325 whole mount embryo has been added (Reviewer1, #1). Panel has been re-arranged for better clarity and the extension of the +325 element is now added in the scheme. Protein IDs in upper case.

Figure 6:

- Panel 6A: locus scheme rearrangement with additional Vista elements (hs1413, hs636, hs1251, hs638). The extension of the VS-250 region is now also indicated.
- Panel 6D: Addition of new qPCR result panels showing *Shox2* levels in WT, *Shox2dc/+* and GDD/*Shox2dc* embryos (Reviewer1, #3).

Figure 7:

- Panel 7A: For completion, DE1 (cranial nerve), DE2 (newly identified craniofacial activity) and DE16 (brain) are now included in the scheme. Cranial nerve, as of neural crest origin and known as a tissue dependent *Shox2* function, is now also listed along with the “neuronal” activity category in the legend. DE11 (activity in blood vessels) has been accidentally listed among the “neuronal enhancers”, while DE14 (hindbrain activity) has not been shown. This is now corrected (DE14 included, DE11 not included).
- Panel 7B: Updates in 7A are included in the scheme. The VS-250 region (dashed dark red line in the “Heart” section) is added to mark the region expected to contain additional cardiac (SV)

enhancers, as indicated by the dashed “oval” with question mark (now placed outside of the scheme).

Supplementary Figures:

Figure S1:

- Panel S1B: The extension of predicted and tested elements and the corresponding conservation tracks have been added for DE2 and DE4 elements.

Figure S2:

- Modified panel S2B: Re-processed ChIP-seq tracks of heart CTCF (at P0, ENCODE) and RAD21/SMC1A (cohesin subunits) from mESCs (Justice et al., 2020; PMID: 32294452) were integrated with the original mESC CTCF track (Reviewer2, #2).

- Figure legend is updated accordingly (including a correction of an accidentally duplicated subtraction analysis paragraph in the previous version).

Figure S3:

- Panel S3B: Following the revision of Fig.1 which now includes the addition of the DE4 element, DE4 is now also marked by a vertical green line to indicate validated limb enhancer-reporter activity. In addition, the GOTHIC scale bars (green, blue and red shades) have been changed from indicating “-log₁₀(q-value)” to “q-value” to more intuitively highlight the calculated interaction strength (scale was shown inverted previously), accordingly also to the scale shown in the new Fig. S5C (v4C from +325 region).

Figure S5:

- Panel S5A: Protein IDs all upper case.

- Panel S5B: Representative +325-LacZ whole mount embryo image has been moved to Fig. 4E (Reviewer1, #1) and was replaced by a close-up for better visibility of cardiac subregional activity in the SV, in direct comparison to the +325B-LacZ transgenic result.

- Panel S5C (NEW): This panel shows the interaction profile of the +325 SV enhancer element across the *Shox2* gene body, gene desert and flanking regions in limb, mandible and heart tissues at E11.5 (setup analogous to the v4C analysis in Fig. S3B). It is the result of the Virtual 4C (v4C) analysis with a +325 element interval as a viewpoint, as requested (Reviewer1, #1).

- Panel S5D: Protein IDs all upper case.

Figure S6:

- New panel S6A: According to the limb epigenomic/chromatin profiling scheme (CTCF, ATAC-seq, H3K27ac ChIP-seq) used in Figure 6A we are now providing close-ups of the different PLE regions allowing more accurate interpretation of peak/signal overlap in context to the respective elements and their extension. This new panel also supports several responses to Reviewer questions, e.g., related to the activity profiles of hs741 and hs1262 (Reviewer1, #4) and PLE4 (Reviewer1, #6), as well as comparison of DE4 and PLE4 enhancer activities.

REVIEWERS' COMMENTS

Reviewer #1 (Remarks to the Author):

This revised version is much improved. The authors have satisfactorily addressed all my previous comments. Their conclusions are fully supported. This is an excellent, very thorough, and interesting paper that deserves to be widely read. I strongly support publication.

Reviewer #2 (Remarks to the Author):

The manuscript is ready for publication

Reviewer #4 (Remarks to the Author):
